# Sensitivity in Translation Averaging

**Lalit Manam**
Indian Institute of Science
Bengaluru, India - 560012
lalitmanam@iisc.ac.in

**Venu Madhav Govindu**
Indian Institute of Science
Bengaluru, India - 560012
venug@iisc.ac.in

## Abstract

In 3D computer vision, translation averaging solves for absolute translations given a set of pairwise relative translation directions. While there has been much work on robustness to outliers and studies on the uniqueness of the solution, this paper deals with a distinctly different problem of sensitivity in translation averaging under uncertainty. We first analyze sensitivity in estimating scales corresponding to relative directions under small perturbations of the relative directions. Then, we formally define the conditioning of the translation averaging problem, which assesses the reliability of estimated translations based solely on the input directions. We give a sufficient criterion to ensure that the problem is well-conditioned. Subsequently, we provide an efficient algorithm to identify and remove combinations of directions which make the problem ill-conditioned while ensuring uniqueness of the solution. We demonstrate the utility of such analysis in global structure-from-motion pipelines for obtaining 3D reconstructions, which reveals the benefits of filtering the ill-conditioned set of directions in translation averaging in terms of reduced translation errors, a higher number of 3D points triangulated and faster convergence of bundle adjustment.

## 1 Introduction

The goal of the translation averaging problem is to recover absolute translations given a redundant set of pairwise relative translation directions. This problem has been relatively less studied compared to the case when pairwise displacements are available [31, 12, 6, 51]. These methods belong to the category of averaging or map synchronization [43, 46], which can be modelled as a network with known pairwise relations between nodes and unknown node values to be estimated. For our specific problem of translation averaging, $\mathcal{G} = (\mathcal{V}, \mathcal{E})$ represents a network, with $N$ nodes, $\mathcal{V}$, denoting the absolute translations $\mathbf{T}_i \in \mathbb{R}^3, i \in \mathcal{V}$, and $M$ edges, $\mathcal{E}$, denoting the pairwise relative direction measurements between the nodes $\mathbf{v}_{ij} \in \mathbb{S}^2, (i, j) \in \mathcal{E}$. Here, all the relative directions $\mathbf{v}_{ij}$'s are in the same coordinate frame. Ideally, $\mathbf{v}_{ij}$ should be the unit vector in the direction of $\mathbf{T}_j - \mathbf{T}_i$. Since the measurements made are directions, these networks are also called bearing-based networks. This problem finds a place in global methods [23, 24, 18, 52] for solving Structure-from-Motion (SfM) [28] in 3D computer vision, where a network of cameras is present with relative translation directions measured between the cameras.

Solving translation averaging is a challenging problem since it requires estimating translation scales in a context of dissimilarity in the input (directions) and output (absolute translations). Due to this dissimilarity, the solutions are defined upto a global scale and a choice of origin. The well-posedness of the bearing-based network problems has been studied under the lens of parallel rigidity theory [45, 19, 36, 42, 56, 2]. This theory determines whether a given set of input directions will have a unique solution (upto a global scale and an origin) for absolute translations.

**Distinction from parallel rigidity and outlier detection:** This paper takes the first step to deal with sensitivity in translation averaging under uncertainty, i.e. determining under what configuration of

37th Conference on Neural Information Processing Systems (NeurIPS 2023).

input directions the solution is reliable. This can be seen as a perturbation analysis of the translation averaging solution with respect to small changes in the input directions. Although there have been empirical studies on small-scale datasets (3 to 20 cameras) on the effect of perturbing input directions [48, 40, 32], to the best of our knowledge, analysis on large-scale networks, both theoretical and empirical, has not been done for bearing-based networks. This analysis is independent of parallel rigidity [2] since parallel rigidity deals with uniqueness of the solution, while we deal with change in solution for small input perturbations. Parallel rigidity can be equivalently described with the algebraic rank of a specific matrix [2], which is similar to the analysis of uniqueness of a solution given a matrix obtained from a linear system of equations. Sensitivity analysis is similar in spirit to the conditioning of a matrix while solving a linear system of equations ($Ax = b$ problem) where the reliability of a solution is studied. Sensitivity is also different from outlier detection in the data because it only deals with small perturbations of the given input directions and not with the noise/outlier levels. The issue of sensitivity remains relevant even without outliers.

**Our contributions:** We first analyze the sensitivity in estimating edge scales in translation averaging by small perturbations to input directions on the smallest solvable bearing-based network. Then, we characterize the conditioning of the translation averaging problem based only on input directions, without any perturbation, for the smallest solvable network i.e. 3 edges between all 3 nodes, and extend it to a general network. We provide a sufficient condition for the network to be well-conditioned. We propose an efficient algorithm to identify ill-conditioned parts of the network, filter them and extract the maximal parallel rigid graph without explicit computation that checks for parallel rigidity. Finally, we show the usefulness of such a filter in the context of SfM with improved translation accuracy, more 3D points triangulated and faster convergence of bundle adjustment after removing ill-conditioned parts of the network.

## 2    Literature Review

In this section, we briefly discuss the relevant literature on translation averaging and parallel rigidity.

**Translation averaging in SfM**: In SfM, the input data contains relative translation directions that are not aligned to a common reference frame. To get the input directions in a common global frame, absolute rotations for each node are estimated using rotation averaging methods [27, 10, 11, 21, 47]. Many translation averaging methods have been proposed over the last two decades. Govindu [23] minimized the cross-product between the input directions and directions obtained from absolute translations. Jiang *et al.* [32] considered triplets of nodes and used the constraints of a triangle to formulate the problem. Wilson *et al.* [55] minimized the deviation between the input directions and directions estimated from absolute translations. Tron *et al.* [54] minimized the squared relative displacements and solved it in a distributed manner. Ozyesil *et al.* proposed the Least Unsquared Deviations (LUD) method, extending [54], with $L_1$ loss for robustness which made the problem as a convex program. Arrigoni *et al.* [5] minimized the squared error of the orthogonal projection of the estimated relative translations onto input directions. Goldstein *et al.* [22] also minimized the orthogonal projection using ADMM but used an $L_1$ loss for robustness. Zhuang *et al.* [57] relaxed the cost in [55] by comparing estimated relative translations to that of the observed directions and called it Bilinear Angle-based Translation Averaging (BATA). Other methods include using two-view and three-view geometry [28] of the cameras to set up the problem [1, 40], estimating edge scales through cycles in a network before solving for absolute translations [3, 4] or through point correspondence constraints [13, 14], iteratively refining input directions [37], averaging matrices obtained from two view geometry [33, 34], and exploiting the structure of the matrix generated from pairwise displacements [15].

**Parallel rigidity**: Several works about parallel rigidity are present in the literature arising from different communities: computer vision [42, 3, 2], robotics [36], computer-aided design [45] and decision control [20, 19, 53, 56]. The node-based formulation is the classical way to approach parallel rigidity, which deals with absolute translations (also called point formation) [45, 20, 19]. The edge-based formulation is a more recent approach which reasons about parallel rigidity based on edge lengths in terms of the cycles in the network [35, 3, 53]. Readers are referred to [2] for an excellent survey on parallel rigidity.

## 3 Sensitivity in Scale Estimation from Directions

In this section, we study the sensitivity in estimating edge scales in translation averaging. At first, we define the notion of a consistent set of directions for a network $\mathcal{G}$, which will be useful for further discussion.

**Definition 1** (Consistent Directions). *A set of relative directions, $\mathbf{v}_{ij}, (i, j) \in \mathcal{E}$ in a bearing-based network $\mathcal{G}$, are said to be consistent if there exist absolute translations, $\mathbf{T}_i, i \in \mathcal{V}$, such that $\frac{\mathbf{T}_j - \mathbf{T}_i}{\|\mathbf{T}_j - \mathbf{T}_i\|} = \mathbf{v}_{ij}$.*

We consider the smallest possible bearing-based network which is solvable, i.e. a network $\mathcal{G}_\Delta$ of 3 nodes, $\mathcal{V}_\Delta = \{1, 2, 3\}$, with all possible edges, $\mathcal{E}_\Delta = \{(1, 2), (2, 3), (3, 1)\}$. When these edges are consistent, they form a triangle. Let $\mathbf{V}$ be the matrix containing relative directions $\mathbf{v}_{ij}, (i, j) \in \mathcal{E}_\Delta$ in its columns and $\mathbf{s}$ be the vector containing edge scales $s_{ij}$. Then, the least squares problem to estimate the scales can be written as

$$\min_{\mathbf{s}} \|\mathbf{V}\mathbf{s}\|^2 \text{ s.t. } \|\mathbf{s}\|^2 = 1, \tag{1}$$

where the unit norm constraint on $\mathbf{s}$ is used to fix the global scale. The solution to the problem in Eqn. 1 is given by the eigenvector corresponding to the smallest eigenvalue of $\mathbf{V}^\mathbf{T}\mathbf{V}$.

To analyze the sensitivity of the estimated scales, we perturb each direction in $\mathcal{G}_\Delta$ by a small 3D rotation $\delta\mathbf{R}_{ij} \in \mathbb{SO}(3)$, therby ensuring that the perturbed vectors always lie on the unit sphere. Let $\mathbf{n}_{ij} \in \mathbb{S}^2$ and $\delta\theta_{ij} > 0$ be the rotation axis and angle for $\delta\mathbf{R}_{ij}$, respectively. Then, the small rotation can be approximated to a first order, using Rodrigues' rotation formula, as $\delta\mathbf{R}_{ij} \approx \mathbf{I} + \delta\theta_{ij}[\mathbf{n}_{ij}]_\times$, where $[\mathbf{n}_{ij}]_\times \in \mathbb{R}^{3\times3}$ is a matrix such that $[\mathbf{n}_{ij}]_\times \mathbf{h} = \mathbf{n}_{ij} \times \mathbf{h}$ for any $\mathbf{h} \in \mathbb{R}^3$, and $\mathbf{I}$ is the $3 \times 3$ identity matrix. For small perturbations to $\mathbf{v}_{ij}$ by $\delta\mathbf{R}_{ij}$, the following theorem holds:

**Theorem 1.** *For a set of consistent directions, $\mathbf{v}_{ij}, (i, j) \in \mathcal{E}_\Delta$, in $\mathcal{G}_\Delta$, the absolute change in any eigenvalue of $\mathbf{V}^\mathbf{T}\mathbf{V}$, denoted as $|\delta\lambda|$, when the directions $\mathbf{v}_{ij}$ are perturbed by small rotations $\delta\mathbf{R}_{ij}$, with $\mathbf{n}_{ij}$ and $\delta\theta_{ij} > 0$ being the rotation axis and angle, is bounded by*

$$|\delta\lambda| \leq \sum_{(k,i,j) \in TI(\Delta)} \delta\theta_{ij} \cdot \|\mathbf{v}_{ki}^{\mathbf{n}_{ij}\perp}\| \cdot \|\mathbf{v}_{kj}^{\mathbf{n}_{ij}\perp}\| \cdot \frac{\left|\sin\phi_{(k,i),(k,j)}^{\mathbf{n}_{ij}\perp}\right|}{\sin^2\phi_{(k,i),(k,j)}} \cdot$$

$$\left[\left(1 + (\mathbf{v}_{ik}^T\mathbf{v}_{jk})^2\right)\left((\mathbf{v}_{ij}^T\mathbf{v}_{ik})^2 + (\mathbf{v}_{ij}^T\mathbf{v}_{jk})^2\right) - 4 \cdot \mathbf{v}_{ik}^T\mathbf{v}_{jk} \cdot \mathbf{v}_{ij}^T\mathbf{v}_{ik} \cdot \mathbf{v}_{ij}^T\mathbf{v}_{jk}\right]^{\frac{1}{2}}, \tag{2}$$

*where $TI(\Delta) = \{(1, 2, 3), (2, 3, 1), (3, 1, 2)\}$, $\mathbf{v}^{\mathbf{n}_{ij}\perp}$ is the component of $\mathbf{v}$ orthogonal to $\mathbf{n}_{ij}$, $\phi_{(k,i),(k,j)}^{\mathbf{n}_{ij}\perp}$ is the angle between $\mathbf{v}_{ki}^{\mathbf{n}_{ij}\perp}$ and $\mathbf{v}_{kj}^{\mathbf{n}_{ij}\perp}$, and $\phi_{(k,i),(k,j)}$ is the angle between $\mathbf{v}_{ki}$ and $\mathbf{v}_{kj}$.*

The proof of Thm. 1 uses eigenvalue perturbation theory [30]. Please refer to the appendix for the proof. From Thm. 1, it can be observed that the bound for absolute change in any eigenvalue is inversely dependent on the square of the sine of angles of the triangle formed by the unperturbed directions. Since the scale estimate, $\mathbf{s}$, is the eigenvector corresponding to the smallest eigenvalue of $\mathbf{V}^\mathbf{T}\mathbf{V}$, Thm. 1 reveals that scales are sensitive to small perturbation of directions when at least one angle in the triangle is small. The term in the square root in Eqn. 2 is bounded due to the dot product of unit norm vectors and is strictly greater than zero since the unperturbed directions are consistent. We note that the bound is also directly proportional to the sine of the angle between the directions projected onto the orthogonal space of the rotation axis. This suggests that the effect of perturbation is maximum when the rotation axis is orthogonal to the directions. This is the same as the case in Cor. 1 when the perturbed directions are consistent.

**Corollary 1.** *For a set of consistent directions, $\mathbf{v}_{ij}, (i, j) \in \mathcal{E}_\Delta$, in $\mathcal{G}_\Delta$, the absolute change in any eigenvalue of $\mathbf{V}^\mathbf{T}\mathbf{V}$, denoted as $|\delta\lambda|$, when the directions $\mathbf{v}_{ij}$ are perturbed by small rotations $\delta\mathbf{R}_{ij}$, with $\mathbf{n}_{ij}$ and $\delta\theta_{ij} > 0$ being the rotation axis and angle, and $\mathbf{n}_{ij}$ being othogonal to $\mathbf{v}_{ki}$ and $\mathbf{v}_{kj}$ for all $(i, j, k) \in TI(\Delta)$, is bounded by*

$$|\delta\lambda| \leq \sum_{(k,i,j) \in TI(\Delta)} \delta\theta_{ij} \cdot \frac{1}{|\sin\phi_{(k,i),(k,j)}|} \cdot$$

$$\left[\left(1 + (\mathbf{v}_{ik}^T\mathbf{v}_{jk})^2\right)\left((\mathbf{v}_{ij}^T\mathbf{v}_{ik})^2 + (\mathbf{v}_{ij}^T\mathbf{v}_{jk})^2\right) - 4 \cdot \mathbf{v}_{ik}^T\mathbf{v}_{jk} \cdot \mathbf{v}_{ij}^T\mathbf{v}_{ik} \cdot \mathbf{v}_{ij}^T\mathbf{v}_{jk}\right]^{\frac{1}{2}}, \tag{3}$$

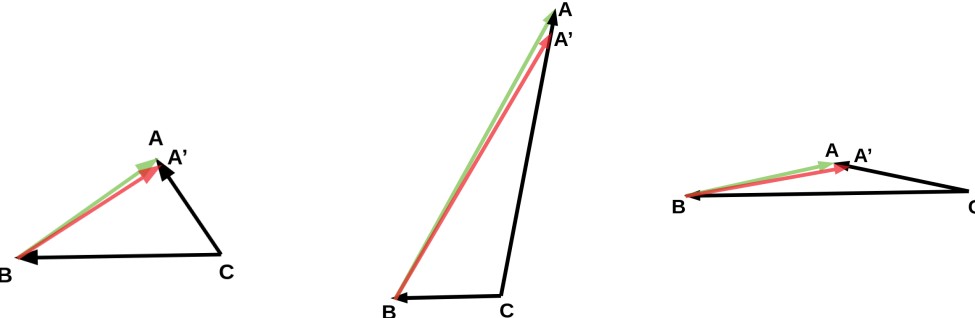

(a) Well-conditioned triangle    (b) Ill-conditioned triangle (Type-I) (c) Ill-conditioned triangle (Type-II)

Figure 1: Conditioning of triangles: analyzing change in output with small perturbation in input direction (green: given direction; red: perturbed direction; change in output: from A to A').

where $TI(\Delta) = \{(1,2,3),(2,3,1),(3,1,2)\}$, and $\phi_{(k,i),(k,j)}$ is the angle between $\mathbf{v}_{ki}$ and $\mathbf{v}_{kj}$.

Cor. 1 reveals that the scale estimates are unstable with small angles, even in the case when the perturbed directions are consistent. We call triangles with at least one small angle as **skewed** triangles. In Fig. 1, we show conditioning of the triangle under different scenarios, with green and red depicting the unperturbed and perturbed directions, respectively. For a well-conditioned triangle, a small change in direction leads to a small change in the absolute translation. But for an ill-conditioned triangle (skewed triangle), a small change in the direction leads to a large change in the absolute translation. Fig. 1a shows a well-conditioned triangle, Fig. 1b shows a triangle with one small angle due to which it is **ill-conditioned (Type I)** and Fig. 1c shows a triangle with two small angles making it **ill-conditioned (Type II)**. Such skewed triangles are known to be problematic in different fields. In numerical analysis [25, 8], the skewed triangles are termed as 'needle-like' triangles, and measuring their area and angles have numerical issues. In computational topology and computer graphics [17, 38], skewed triangles (also called sliver triangles) are often avoided, as in Delaunay triangulation, or are filtered out. We note that none of them deal with directions as input. Our aim is to quantify the conditioning of the triangle and the translation averaging problem based solely on the input directions, without actual perturbation, and filter out skewed triangles, which are discussed in subsequent sections.

## 4   Conditioning of Translation Averaging

In this section, we first analyze the conditioning of the bearing-based network with 3 nodes and 3 edges, i.e. $\mathcal{G}_\Delta$ (as defined in Sec. 3). We define an *angle matrix*, $\mathbf{A}_{\mathcal{G}_\Delta} \in \mathbb{R}^{3\times3}$, as follows:

$$\mathbf{A}_{\mathcal{G}_\Delta} = \begin{bmatrix} \phi_{(1,2),(1,2)} & \phi_{(2,1),(2,3)} & \phi_{(1,2),(1,3)} \\ \phi_{(2,3),(2,1)} & \phi_{(2,3),(2,3)} & \phi_{(3,2),(3,1)} \\ \phi_{(1,3),(1,2)} & \phi_{(3,1),(3,2)} & \phi_{(3,1),(3,1)} \end{bmatrix} = \begin{bmatrix} 0 & \phi_{(2,1),(2,3)} & \phi_{(1,2),(1,3)} \\ \phi_{(2,1),(2,3)} & 0 & \phi_{(3,2),(3,1)} \\ \phi_{(1,2),(1,3)} & \phi_{(3,2),(3,1)} & 0 \end{bmatrix}, \quad (4)$$

where $\phi_{(k,i),(k,j)}$ is the angle between $\mathbf{v}_{ki}$ and $\mathbf{v}_{kj}$ (as defined in Thm. 1). The rows and columns of $\mathbf{A}_{\mathcal{G}_\Delta}$ signify edges in $\mathcal{G}_\Delta$ and its entries are the angular differences between directions with angles being measured within the triangle using the common node between two edges as the reference point for measuring directions. Since $\phi_{(k,i),(k,j)} = \phi_{(k,j),(k,i)}$, $\mathbf{A}_{\mathcal{G}_\Delta}$ is symmetric, and since $\phi_{(k,i),(k,i)} = 0$, $\mathbf{A}_{\mathcal{G}_\Delta}$ has zero diagonal entries. We quantify conditioning of the translation averaging problem on $\mathcal{G}_\Delta$ as conditioning of the matrix $\mathbf{A}_{\mathcal{G}_\Delta}$ based on the following theorem:

**Theorem 2.** *Consider the bearing based-network of 3 nodes and 3 edges, $\mathcal{G}_\Delta = (\mathcal{V}_\Delta, \mathcal{E}_\Delta)$, and the corresponding angle matrix $\mathbf{A}_{\mathcal{G}_\Delta}$. The conditioning of the matrix $\mathbf{A}_{\mathcal{G}_\Delta}$ signifies the skewness of the triangle formed using the directions in $\mathcal{E}_\Delta$.*

The proof of Thm. 2 involves proving $\mathbf{A}_{\mathcal{G}_\Delta}$ being non-singular for non-zero angles using its determinant and checking for the closeness of columns in relation to the angle values. We choose this approach instead of checking determinant since the later is not a good measure of closeness of a matrix to singularity [39]. Please refer to the appendix for the complete proof. Thm. 2 reveals that the conditioning of $\mathcal{G}_\Delta$ can be characterized by the condition number of matrix $\mathbf{A}_{\mathcal{G}_\Delta}$.

We generalize the angle matrix and Thm. 2 for any bearing-based network $\mathcal{G}$. We assume all edges in $\mathcal{G}$ are a part of at least one **triplet** (a triplet consists of 3 nodes with all possible edges between them). We use the terms **triplet** and **triangle** interchangeably since a triplet forms a triangle for our specific problem. If this assumption is not satisfied, we remove the edges in $\mathcal{G}$, which are not a part of any triplet and consider its maximum connected component. This will enable us to use $\mathcal{G}_\triangle$ as the basic building block of $\mathcal{G}$ and thus characterizing the conditioning of $\mathcal{G}$.

We now expand the angle matrix to a general bearing-based network $\mathcal{G}$ in which every edge is a part of at least one triplet. We denote the angle matrix for the general network as $\mathbf{A}_\mathcal{G} \in \mathbb{R}^{M \times M}$ ($M$ is the number of edges in $\mathcal{G}$), where the rows and columns signify edges in $\mathcal{G}$, and its entries, $a_\mathcal{G}^{ij,kl}$, corresponding to the row for the edge $(i,j)$ and the column for the edge $(k,l)$, are defined as follows:

$$a_\mathcal{G}^{ij,kl} = \begin{cases} \phi_{(c,c_1'),(c,c_2')} & \text{if } (i,j) \text{ and } (k,l) \text{ belong to the same triplet} \\ & \text{with } c \in \{i,j\} \cap \{k,l\} \text{ and } c_1', c_2' \in \{i,j\} \cup \{k,l\} \setminus \{c\}; \\ 0 & \text{otherwise.} \end{cases} \quad (5)$$

In $\mathbf{A}_\mathcal{G}$, we measure the angles between the directions formed by triplet. In such cases, to ensure that the directions are measured within the triangle, similar to $\mathbf{A}_{\mathcal{G}_\triangle}$, the common node, $c$, is taken as reference and the directions are measured from $c$ to other nodes $c_1'$ and $c_2'$. From Eqn. 5, it is clear that the diagonal entries are zero ($\phi_{(k,i),(k,i)} = 0$) and the matrix is symmetric ($\phi_{(k,i),(k,j)} = \phi_{(k,j),(k,i)}$). We note that $\mathbf{A}_\mathcal{G}$ is not the same as that of the distance matrix used in Multi-Dimensional Scaling (MDS) [7, 9]. In instances when a node appears in more than one triplet, angles between the edges belonging to the different triplets are not computed, but in MDS, such angles would also be considered. This restricts the angle matrix $\mathbf{A}_\mathcal{G}$ to contain information only about triangles and not of other structures. Such a construction helps to extend the findings from $\mathbf{A}_{\mathcal{G}_\triangle}$ to $\mathbf{A}_\mathcal{G}$. We define the **conditioning of the translation averaging problem** as the condition number of angle matrix $\mathbf{A}_\mathcal{G}$. The following theorem states a sufficiency condition for a well-conditioned angle matrix:

**Theorem 3.** *Consider a bearing-based network $\mathcal{G}$, with all edges contributing to triplets. The angle matrix $\mathbf{A}_\mathcal{G}$, corresponding to $\mathcal{G}$, is well conditioned if the minimum angle (or equivalently all the angles) in all the triangles formed by the triplets are sufficiently large.*

The proof for 3 is similar to the proof for Thm. 2. Here, we check for different combinations of columns based on the structure of $\mathcal{G}$ and check for their closeness. Please refer to the appendix for the proof.

We note that we do not need $\mathcal{G}$ to be parallel rigid for the angle matrix $\mathbf{A}_\mathcal{G}$ to be well conditioned, implying that sensitivity and parallel rigidity are different aspects of the bearing-based network. However, we need $\mathcal{G}$ to be parallel rigid to ensure a unique solution. In general, computing the maximal parallel rigid component is expensive. Since we remove edges in $\mathcal{G}$ not contributing to triplets, this can affect its parallel rigidity. We avoid such expensive computation since we deal with triplets. We construct a **triplet network** $\mathcal{G}_T = (\mathcal{V}_T, \mathcal{E}_T)$, as done in [32], where nodes $\mathcal{V}_T$ denote a triplet in $\mathcal{G}$ and edges $\mathcal{E}_T$ connect the nodes if an edge is common between the triplets in $\mathcal{G}$. By construction, it can be seen that disconnected components in $\mathcal{G}$ will be disconnected in $\mathcal{G}_T$ since there cannot be shared edges among disconnected components. The following theorem enables extracting the maximal parallel rigid graph when $\mathcal{G}$ contains all edges contributing to triplets:

**Theorem 4.** *Given a bearing-based network $\mathcal{G}$, with all edges contributing to triplets forming triangles, and its corresponding triplet network $\mathcal{G}_T$, the maximal parallel rigid component of $\mathcal{G}$ can be determined by the edges in $\mathcal{G}$ contributing to the largest connected component of $\mathcal{G}_T$.*

Please refer to the appendix for proof of Thm. 4. Based on Thms. 3 and 4, we develop an algorithm to identify and remove skewed triangles from the translation averaging problem while ensuring that the network is parallel rigid, which is discussed in the next section.

## 5   Proposed Method

In this section, we show how to efficiently identify skewed triangles in $\mathcal{G}$. First, we get the list of nodes and edges contributing to each triplet in $\mathcal{G}$, for which efficient implementations exist [52, 26]. The brute force way to identify skewed triangles is to compute the angles in each triangle using the relative directions $\mathbf{v}_{ij}$ and mark triangles with the minimum angle less than a threshold. The number

of triplets depends on the sparsity of the network $\mathcal{G}$, which, in general, is large. This makes the brute force method time-consuming. In contrast with the brute force approach, we use vectorized operations to construct the angle matrix $\mathbf{A}_\mathcal{G}$ and filter skewed triangles, making it time efficient. It is observed that the vectorized version is $\sim 100$ times faster than the parallelized version (with 20 threads) of the brute force method for $\sim 10^5$ triplets coming from $\sim 10^4$ edges, and this gain increases significantly with an increase in the number of triplets and edges. A time comparison between the brute force method and our method is given in the appendix. In real-world data, the large-scale bearing-based networks are sparse, making $\mathbf{A}_\mathcal{G}$ sparse. We present our method for sparse networks in the following steps and for dense networks in the appendix.

**Step 1:** The first step is to identify which entries are non-zero in the angle matrix $\mathbf{A}_\mathcal{G}$. Since we are considering only triplets, we require a matrix that captures which edges are part of a triplet. In computational topology [29, 16], higher order relationships in $\mathcal{G} = (\mathcal{V}, \mathcal{E})$ (other than node-to-node relationship via edges) are studied using simplices. A $k$-simplex is a subset of the vertex set $\mathcal{V}$ with $(k + 1)$ elements. A finite collection of simplices, such as nodes (0-simplices), edges (1-simplices), and triangles (2-simplices), is called a simplicial complex. The boundary matrix $\mathbf{B}_k$ of a simplicial complex encodes which $(k - 1)$-simplex contributes to $k$-simplex. In our case, we need the relationship between 1-simplices (edges) and 2-simplices (triangles). The $(i, j)^{th}$ element of the boundary matrix $\mathbf{B}_2 \in \mathbb{R}^{M \times W}$, with $M$ edges and $W$ triplets, is given as:

$$b_2^{ij} = \begin{cases} 1 & \text{if } i^{th} \text{ edge contributes to } j^{th} \text{ triangle;} \\ 0 & \text{otherwise.} \end{cases} \tag{6}$$

It is easy to compute $\mathbf{B}_2$ since we have the list of edges contributing to each triplet. The row and column indices of non-zero elements in $\mathbf{B}_2 \mathbf{B}_2^T$ give us the edge pairs participating in triplets. Only these elements in $\mathbf{A}_\mathcal{G}$ are to be computed (except diagonals in $\mathbf{B}_2 \mathbf{B}_2^T$ which are non-zero). The corresponding dot products between the edges are computed in a vectorized fashion and stored.

**Step 2:** We need to identify the correct signs for the dot products to ensure that the dot products reflect the cosine of the angles of the triangles. Since the dot products are $\cos \phi_{(i,j),(k,l)}$ for edges $(i, j), (k, l) \in \mathcal{E}$ in triplets, there will be one common node $c \in \{i, j\} \cap \{k, l\}$. The directions $\mathbf{v}_{ij}$ and $\mathbf{v}_{kl}$ should be measured with reference to the common node to get the angle of the triangle that includes those edges. We need the dot product between $\mathbf{v}_{cc_1'}$ and $\mathbf{v}_{cc_2'}$, where $c_1' \in \{i, j\} \setminus \{c\}$ and $c_2' \in \{k, l\} \setminus \{c\}$. Using $\mathbf{v}_{cc_1'}^T \mathbf{v}_{cc_2'} = \mathbf{v}_{c_1'c}^T \mathbf{v}_{c_2'c}$ and $\mathbf{v}_{ij} = -\mathbf{v}_{ji}$, we premultiply each dot product by $m^{ij,kl}$, where $m^{ij,kl}$ is determined as follows:

$$m^{ij,kl} = \begin{cases} 1 & \text{if } i = k \text{ or } j = l; \\ -1 & \text{if } i = l \text{ or } j = k; \\ 0 & \text{otherwise.} \end{cases} \tag{7}$$

Since edges $(i, j), (k, l)$ are part of a triplet, the third case in Eqn. 7 will not arise, but it will be helpful in the dense network case. We premultiply $m^{ij,kl}$ to the corresponding dot products. We take the inverse cosine of the sign-corrected dot products to get the angles and allocate $\mathbf{A}_\mathcal{G}$.

**Step 3:** It is easy to extract angles from $\mathbf{A}_\mathcal{G}$ since edges corresponding to a triplet are known and $\mathbf{A}_\mathcal{G}$ provides an easy way to index edges based on its rows and columns. Let $Trp$ be the set of triplets in $\mathcal{G}$. We check if the angles of the triangles are greater than a minimum threshold and remove the triplets having skewed triangles. This creates two sets of triplets $Trp_{Ret}$ and $Trp_{Rem}$ denoting retained and removed triplets, both being mutually exclusive. Let the edges contributing to the two triplet sets be $\mathcal{E}_{ret}$ and $\mathcal{E}_{rem}$, respectively, such that $\mathcal{E} = \mathcal{E}_{ret} \cup \mathcal{E}_{rem}$. We note that $\mathcal{E}_{ret} \cap \mathcal{E}_{rem}$ is not necessarily empty. For a parallel rigid $\mathcal{G}$, the two edge sets are not mutually exclusive since there will be common edges between the triplets of $\mathcal{E}_{ret}$ and $\mathcal{E}_{rem}$, as a consequence of Thm. 4. Let the filtered network be denoted as $\tilde{\mathcal{G}}_F = (\tilde{\mathcal{V}}_F, \tilde{\mathcal{E}}_F)$. There are two ways to remove edges: one is to remove edges **aggressively** such that $\tilde{\mathcal{E}}_F = \mathcal{E} \setminus \mathcal{E}_{rem}$ and the other way is **non-aggressive** i.e. $\tilde{\mathcal{E}}_F = \mathcal{E}_{ret}$. Since all operations are vectorized, this makes the whole process of extracting the angles and removal of skewed triangles time efficient. Next, we construct the triplet network $\mathcal{G}_T$ (as discussed in Sec. 4) from $\tilde{\mathcal{G}}_F$ and extract the largest connected component of $\mathcal{G}_T$, to ensure that the network is connected and is parallel rigid. The edges of $\tilde{\mathcal{G}}_F$ contributing to the largest connected component of $\mathcal{G}_T$ gives the final network $\mathcal{G}_F = (\mathcal{V}_F, \mathcal{E}_F)$. The whole process is summarized in Algo 1. In our experiments, the network $\mathcal{G}$ is sparse, and we choose the non-aggressive way of removing the edges, which ensures

| **Algorithm 1:** Removal of Skewed Triangles from Sparse Networks |
|---|

**Input:** Bearing-based network $\mathcal{G} = (\mathcal{V}, \mathcal{E})$, containing only triplets and is parallel rigid, and the triplet list of $\mathcal{G}$.

**Output:** Bearing-based network $\mathcal{G}_F = (\mathcal{V}_F, \mathcal{E}_F)$, containing only triplets and is parallel rigid, without skewed triangles.

1 Compute the non-zero elements in angle matrix $\mathbf{A}_{\mathcal{G}}$ using the boundary matrix $\mathbf{B}_2$ (Eqn. 6).
2 Compute the dot product for the non-zero elements in $\mathbf{A}_{\mathcal{G}}$.
3 Get the signs of the dot product i.e. $m^{ij,kl}$ using Eqn. 7.
4 Multiply dot products with their corresponding signs $m^{ij,kl}$.
5 Take the inverse cosine of dot products.
6 Allocate the matrix $\mathbf{A}_{\mathcal{G}}$. Extract the angles of the triplets from $\mathbf{A}_{\mathcal{G}}$.
7 Filter out the triplets with the minimum angle less than a threshold to get $\tilde{\mathcal{G}}_F = (\tilde{\mathcal{V}}_F, \tilde{\mathcal{E}}_F)$.
8 Construct the triplet network $\mathcal{G}_T$ using $\tilde{\mathcal{G}}_F$.
9 Get the largest connected component of $\mathcal{G}_T$.
10 Get the edges of $\tilde{\mathcal{G}}_F$ contributing to the largest connected component of $\mathcal{G}_T$ to get $\mathcal{G}_F = (\mathcal{V}_F, \mathcal{E}_F)$, which is connected and is parallel rigid.

that the connected component of $\mathcal{G}_F$ is large. We note that with non-aggressive pruning, we still ensure that the nodes are estimated reliably since the nodes are also a part of atleast one triplet, which belong to the set of non-skewed triangles.

## 6 Experiments

We consider SfM datasets provided in 1DSfM [55] for the experiments. It provides the relative motions and a reference reconstruction using Bundler [49, 50]. Since Bundler was published more than ten years ago, we use COLMAP [44] to generate the pairwise relative rotations and translations and use COLMAP's solution as the ground truth to get a better reconstruction. We take COLMAP solution and align it to the solution provided in 1DSfM to get absolute translations in meters. We compute absolute rotations using [11] and use them to align the relative translation directions into a global coordinate frame. The output of this process gives us a bearing-based network. Then, we extract triplets from the network and ensure parallel rigidity. Since the filtering process is independent of the cost function to solve the problem, we consider two representative cost functions. Revised LUD (an improvement on [41], provided by [57]), compares relative displacements, and BATA [57], compares relative directions. Please refer to appendix for the problem formulations. Our code is implemented in MATLAB. All experiments are performed on a PC with Intel Xeon Silver 4210 processor with 128 GB RAM. Finally, in each table, **Mean-ATE** and **RMS-ATE** denote the **mean** and **RMS** absolute translation errors, respectively, **w/o** and **w/ filter** denotes the network **before** and **after removing** skewed triangles and **bold** entries denote better performance between the solutions of the two networks.

### 6.1 Analysis of Real Data

In this subsection, we first provide an illustrative example using COLMAP [44] reconstruction and then analyze the real data. Fig. 2 shows an illustrative example of the existence of such skewed triangles in the SfM problem. For the Alamo dataset, most images capture the front part of the museum, and thus, the cameras are densely connected in the network. The blue triangle depicts a triplet which is Type-I ill-conditioned triangle and the green triangle shows Type-II ill-conditioned triangle.

Now, we analyze the real data to understand the frequency of occurrence of skewed triangles and their difference with outlier data. For the analysis, we compute the errors in relative directions with respect to ground truth. To check whether atleast one outlier is present in the triplet, we check the maximum error of relative directions in a triplet. Since we identify skewed triangles with the minimum angle in each triplet, we compare the maximum error of relative directions with the minimum angle between directions for each triplet. In Fig. 3, we show scatter plots between the two compared quantities for all the triplets for three datasets with different edge densities. At first, we observe that there are a

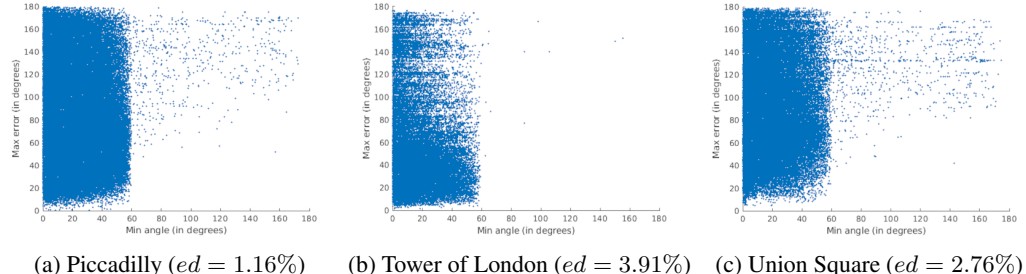

(a) Piccadilly ($ed = 1.16\%$)     (b) Tower of London ($ed = 3.91\%$)     (c) Union Square ($ed = 2.76\%$)

Figure 3: Scatter plots of maximum error of relative directions in each triplet (with respect to ground truth) vs minimum angle between edges in each triplet on datasets from [55]. $ed$ denotes edge density.

considerable number of skewed triangles in the network since there are many points in the scatter plot which are close to $0°$ in the x-axis (representing the minimum angle in triplets). Next, we can also see that the minimum angle between edges in the triplets is independent of the maximum error in the triplets, implying that the presence of an outlier and the skewness of a triangle have no relation. Also, it can be seen that triplets with minimum angle $> 60°$ are clear outliers and all of them have a high angular error with respect to ground truth. Based on this observation, we first provide results on outlier-free data, and then we show the results on the real data with outliers in the next subsections.

## 6.2 Outlier-Free Data

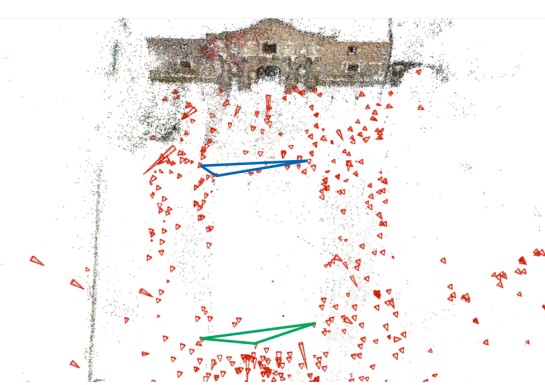

Figure 2: Reference reconstruction of Alamo [55] using COLMAP [44] for displaying skewed triangles. Blue triangle: Ill-conditioned triangle (Type-I), Green triangle: Ill-conditioned triangle (Type-II).

In this subsection, we examine the impact of skewed triangles on the quality of the translation estimate independent of outliers. First, we extract the component of the network for which the ground truth is available. For this experiment, to remove the effect of errors coming from absolute rotation estimates, we use ground truth rotations to align the relative directions to a common reference frame. We consider the edges as outliers if the relative direction on the edge differs from its ground truth equivalent by more than $10°$ and remove them. Then, we extract the triplets from the network and ensure parallel rigidity. This gives us a network with no outliers but still contains skewed triangles. We call it the unfiltered network. Now, we use Algo. 1 to remove skewed triangles (minimum angle $< 5°$) and denote the output as the filtered network and compare the solutions from the two networks.

In Table 1, we list the number of nodes and edges removed due to the removal of skewed triangles and check the absolute translation errors obtained using BATA. It can be seen with the removal of a small number of nodes and edges, the mean and RMS errors of the absolute translations decrease consistently (see appendix for the number of nodes and edges of the networks). We also check the errors of the nodes in the unfiltered network, which were removed due to the removal of skewed triangles. It can be seen that the mean and RMS errors of the removed nodes are high compared to the overall errors implying that absolute translations at these nodes are not well estimated. Absolute translation error obtained using Revised LUD is provided in the appendix, which also shows that removal of skewed triangles leads to improved translation estimates. This shows the impact of skewed triangles on the translation averaging and the benefits of removing them.

Table 1: Absolute translations errors (in meters) on 1DSfM [55] datasets without outliers using BATA [57]. Removed Node Errors: Errors of removed nodes in the unfiltered network, $\#N_{rem}, \#M_{rem}$: No. of nodes and edges removed.

| Dataset | $\#N_{rem}$ | $\#M_{rem}$ | Mean-ATE | | RMS-ATE | | Removed Node Errors | |
|---|---|---|---|---|---|---|---|---|
| | | | w/o filter | w/ filter | w/o filter | w/ filter | Mean | RMS |
| Alamo (ALM) | 17 | 211 | 2.4 | **2.2** | 4.4 | **3.9** | 8.0 | 10.0 |
| Ellis Island (ELS) | 1 | 15 | 1.0 | **0.9** | 1.5 | **1.3** | 5.5 | 5.5 |
| Gendarmenmarkt (GMM) | 9 | 67 | 5.1 | **4.8** | 8.5 | **7.9** | 19.2 | 21.8 |
| Madrid Metropolis (MDR) | 20 | 184 | 7.0 | **6.0** | 13.6 | **10.6** | 23.6 | 29.7 |
| Montreal Notre Dame (MND) | 6 | 163 | 2.4 | **2.3** | 4.1 | **3.8** | 7.1 | 8.9 |
| NYC Library (NYC) | 20 | 185 | 2.5 | **2.3** | 5.1 | **5.0** | 6.2 | 8.2 |
| Notre Dame (ND) | 17 | 536 | 2.5 | **2.4** | 5.0 | **4.8** | 7.6 | 11.8 |
| Piazza del Popolo (PDP) | 10 | 247 | 2.7 | **2.6** | 4.2 | 4.5 | 4.3 | 5.6 |
| Piccadilly (PIC) | 49 | 780 | 2.3 | **2.1** | 5.4 | **4.6** | 8.6 | 17.0 |
| Roman Forum (ROF) | 30 | 391 | 9.2 | **8.9** | 19.1 | **16.7** | 26.9 | 42.9 |
| Tower of London (TOL) | 11 | 96 | 8.0 | **7.5** | 16.2 | **14.3** | 37.8 | 53.5 |
| Trafalgar (TFG) | 184 | 1879 | 8.0 | **6.9** | 17.1 | **17.0** | 21.7 | 27.9 |
| Union Square (USQ) | 21 | 239 | 4.9 | **4.5** | 6.7 | **6.0** | 10.6 | 13.2 |
| Vienna Cathedral (VNC) | 19 | 275 | **7.2** | 7.4 | 10.6 | **10.4** | 15.0 | 18.0 |
| Yorkminster (YKM) | 13 | 181 | 4.9 | **4.8** | **13.4** | 14.8 | 17.5 | 26.5 |

Table 2: Details of networks before and after removing skewed triangles from 1DSfM [55] datasets. $\#N_{rem}, \#M_{rem}$: No. of nodes and edges removed, $\kappa_2(\mathbf{A}_{\mathcal{G}})$: condition number of the angle matrix with matrix-2 norm, $t_{filter}$: time taken to remove skewed triangles.

| Dataset | #Nodes | | #Edges | | $\#N_{rem}$ | $\#M_{rem}$ | $\kappa_2(\mathbf{A}_{\mathcal{G}})$ | | $t_{filter}$ (sec) |
|---|---|---|---|---|---|---|---|---|---|
| | w/o filter | w/ filter | w/o filter | w/ filter | | | w/o filter | w/ filter | |
| ALM | 694 | 682 | 15619 | 15422 | 12 | 197 | 2.3e+08 | 2.4e+07 | 0.31 |
| ELS | 324 | 316 | 7411 | 7342 | 8 | 69 | 2.6e+07 | 3.8e+06 | 0.14 |
| GMM | 950 | 926 | 13913 | 13656 | 24 | 257 | 2.7e+09 | 1.8e+08 | 0.20 |
| MDR | 401 | 377 | 4367 | 4170 | 24 | 197 | 7.2e+07 | 5.6e+06 | 0.05 |
| MND | 564 | 560 | 18297 | 18150 | 4 | 147 | 2.1e+08 | 3.4e+07 | 0.67 |
| NYC | 450 | 437 | 6228 | 6017 | 13 | 211 | 6.0e+07 | 4.0e+06 | 0.08 |
| ND | 1421 | 1418 | 70759 | 70651 | 3 | 108 | 5.4e+09 | 1.2e+10 | 2.66 |
| PDP | 909 | 891 | 14770 | 14502 | 18 | 268 | 1.3e+08 | 3.1e+07 | 0.32 |
| PIC | 2706 | 2645 | 45306 | 44405 | 61 | 901 | 1.7e+09 | 6.5e+08 | 0.55 |
| ROF | 1361 | 1320 | 18855 | 18461 | 41 | 394 | 2.5e+09 | 4.9e+08 | 0.25 |
| TOL | 569 | 552 | 8710 | 8569 | 17 | 141 | 8.2e+07 | 2.0e+07 | 0.14 |
| TFG | 6327 | 6110 | 110876 | 108142 | 217 | 2734 | 1.5e+11 | 4.7e+09 | 1.83 |
| USQ | 866 | 840 | 12638 | 12406 | 26 | 232 | 2.7e+09 | 2.6e+07 | 0.23 |
| VNC | 1015 | 986 | 24793 | 24511 | 29 | 282 | 1.3e+09 | 1.2e+09 | 0.65 |
| YKM | 964 | 930 | 12033 | 11670 | 34 | 363 | 1.0e+12 | 1.0e+12 | 0.18 |

## 6.3 Real Data

In this subsection, we deal with real data in the context of SfM, which is obtained as described at the beginning of this section. We note that this data contain outliers, and our aim is to understand the impact of skewed triangles in the presence of outliers. We denote the network obtained as unfiltered and remove the skewed triangles (minimum angle $< 5°$) using Algo. 1, calling it the filtered network. In Table 2, we provide details of the unfiltered and filtered network. It can be observed that the removal of a small fraction of nodes and edges from skewed triangles leads to a significant decrease in the condition number of the angle matrix (with matrix-2 norm), $\kappa_2(\mathbf{A}_{\mathcal{G}})$, for all datasets expect ND. We employ non-aggressive pruning due to which some skewed triangles can remain in the filtered network making the condition number $\kappa_2(\mathbf{A}_{\mathcal{G}})$ increase for ND and the same for YKM. We reiterate that non-aggressive pruning still ensures that the nodes are well estimated (see Step 3 in Sec. 5). This shows that the conditioning of the translation averaging improves for most datasets, even with the non-aggressive removal of skewed triangles.

In Table 3, we present the absolute translation errors obtained using BATA without and with filtering skewed triangles. It can be observed that the mean and RMS errors improve for most of the datasets after the removal of skewed triangles. Also, errors of the removed nodes in the unfiltered network are high, which indicates that the absolute translations at these nodes are not reliable in the unfiltered network. We note that from Tables 2 and 3, the filtering time is $\sim 1\%$ of the time taken for translation averaging, which shows the practicality of Algo. 1. Next, we perform 3D reconstruction using

Table 3: Absolute translations errors (in meters) on 1DSfM [55] datasets using BATA [57]. Removed Node Errors: Errors of removed nodes in the unfiltered network, $t_{BATA}$: time taken by BATA.

| Dataset | Mean-ATE | | RMS-ATE | | Removed Node Errors | | $t_{BATA}$ (sec) |
|---|---|---|---|---|---|---|---|
| | w/o filter | w/ filter | w/o filter | w/ filter | Mean | RMS | |
| ALM | 4.7 | **4.5** | 11.1 | **10.5** | 22.9 | 39.5 | 17 |
| ELS | 23.2 | **22.1** | **50.7** | 51.8 | 98.6 | 118.1 | 7 |
| GMM | 50.6 | **40.9** | 77.8 | **61.1** | 149.7 | 190.8 | 14 |
| MDR | 13.9 | **12.7** | 29.4 | **26.1** | 54.2 | 61.2 | 5 |
| MND | 4.4 | **4.3** | 10.0 | **9.9** | 31.2 | 34.1 | 17 |
| NYC | 6.5 | **5.2** | 15.8 | **12.6** | 28.7 | 38.1 | 7 |
| ND | **3.3** | **3.3** | **6.4** | **6.4** | 6.9 | 7.0 | 71 |
| PDP | **8.0** | **8.0** | **13.0** | 13.1 | 15.0 | 17.3 | 16 |
| PIC | 5.3 | **5.1** | 11.0 | **10.8** | 20.8 | 29.2 | 68 |
| ROF | 12.8 | **10.4** | 27.0 | **19.6** | 65.4 | 100.5 | 24 |
| TOL | 15.6 | **14.5** | 32.2 | **30.3** | 63.7 | 90.3 | 10 |
| TFG | 20.3 | **14.5** | 62.6 | **31.8** | 65.0 | 130.8 | 287 |
| USQ | 14.5 | **10.6** | 24.8 | **18.1** | 34.5 | 49.9 | 14 |
| VNC | 10.2 | **10.2** | **17.8** | 18.4 | 21.4 | 27.5 | 28 |
| YKM | 20.4 | **19.3** | 29.5 | **28.3** | 44.0 | 51.4 | 13 |

Table 4: No. of points triangulated ($P_{tri} \times 10^3$) and bundle adjustment iterations ($BA_{iters}$) using the solutions obtained by BATA for unfiltered and filtered networks on 1DSfM [55] datasets.

| Dataset | | ALM | ELS | GMM | MDR | MND | NYC | ND |
|---|---|---|---|---|---|---|---|---|
| $P_{tri} \uparrow$ | w/o filter | 142 | 57 | 101 | 46 | 115 | 77 | **376** |
| | w/ filter | **144** | **58** | **104** | **49** | **118** | **79** | 375 |
| $BA_{iters} \downarrow$ | w/o filter | 73 | 40 | 31 | **66** | 59 | 100 | 34 |
| | w/ filter | **54** | **33** | **21** | 83 | **37** | **74** | **22** |
| | | PDP | PIC | ROF | TOL | USQ | VNC | YKM |
| $P_{tri} \uparrow$ | w/o filter | 91 | 231 | **196** | 107 | **57** | 219 | 175 |
| | w/ filter | **94** | **234** | **196** | **110** | 56 | **222** | **168** |
| $BA_{iters} \downarrow$ | w/o filter | **34** | 53 | 31 | 89 | 122 | 52 | 58 |
| | w/ filter | 44 | **47** | **21** | **85** | **52** | **25** | **33** |

Theia [52] with the absolute translation solutions obtained in Table 3. Theia removes the triangulated 3D points which have reprojection errors greater than 15 pixels. It can be seen from Table 4 that more 3D points are triangulated in the filtered networks for most of the datasets. We note that filtered networks have lesser nodes than unfiltered networks, which implies removal of nodes coming from skewed triangles leads to better conditioning for 3D point triangulation. We also see that the bundle adjustment converges faster for the filtered networks indicating absolute translations are more stable in the filtered networks compared to unfiltered networks.

**Limitation:** Our sensitivity analysis of translation averaging is based on the triplets in the network. Although we do not lose much in terms of 3D reconstruction in SfM, further work is required to understand the sensitivity of a general bearing-based network.

# 7   Conclusion

This paper deals with sensitivity in translation averaging under input uncertainty. We study sensitivity in estimating edge scales in bearing-based networks which suggests skewed triangles are unstable. We define the conditioning of the translation averaging problem and provide a sufficient criterion to ensure that the problem is well-conditioned. Then, we propose an efficient algorithm to remove skewed triangles from the network while ensuring parallel rigidity. We demonstrate the effectiveness of our filtering scheme using structure-from-motion data without and with outliers leading to better absolute translation estimates, more 3D points triangulated and faster convergence of bundle adjustment for filtered networks.

## Acknowledgments and Disclosure of Funding

Lalit Manam is supported by a Prime Minister's Research Fellowship, Government of India. This research was supported in part by a Core Research Grant from Science and Engineering Research Board, Department of Science and Technology, Government of India.

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

# Sensitivity in Translation Averaging
# Appendix

**Lalit Manam**
Indian Institute of Science
Bengaluru, India - 560012
lalitmanam@iisc.ac.in

**Venu Madhav Govindu**
Indian Institute of Science
Bengaluru, India - 560012
venug@iisc.ac.in

## 8   Proofs for Theorems

Before we prove the theorems, we first state the results, which will be helpful in proving Thm. 1. We first state a result from first-order eigenvalue perturbation theory [30].

**Result 1.** *Given a matrix $\mathbf{X} \in \mathbb{R}^{M \times M}$ and a small additive perturbation on it by a matrix $\delta\mathbf{X} \in \mathbb{R}^{M \times M}$, the absolute change in eigenvalue, denoted as $|\delta\lambda(\mathbf{X})|$, due to the perturbation is bounded by*

$$|\delta\lambda(\mathbf{X})| \leq \frac{\|\mathbf{w}\|_2 \|\delta\mathbf{X}\|_2 \|\mathbf{z}\|_2}{|\mathbf{w}^H \mathbf{z}|}, \tag{8}$$

*where $\mathbf{w}, \mathbf{z} \in \mathbb{C}^M$, and are the left and right eigenvectors of $\mathbf{X}$ corresponding to the same eigenvalue, respectively.*

Since $\mathbf{w}$ and $\mathbf{z}$ are eigenvectors, $\|\mathbf{w}\|_2 = 1$ and $\|\mathbf{z}\|_2 = 1$. For a symmetric matrix $\mathbf{X}$, the left and the right eigenvectors corresponding to any eigenvalue are the same due to which $|\mathbf{w}^H \mathbf{z}| = |\mathbf{w}^H \mathbf{w}| = 1$. Hence, for symmetric matrix $\mathbf{X}$, Result 1 can be rewritten as

$$|\delta\lambda(\mathbf{X})| \leq \|\delta\mathbf{X}\|_2 = \sigma_{max}(\delta\mathbf{X}), \tag{9}$$

where $\sigma_{max}(\delta\mathbf{X})$ is the maximum singular value of $\delta\mathbf{X}$. Next, we provide our result on the singular value decomposition of a specific matrix, which will be used further.

**Result 2.** *Given a matrix $\mathbf{X} \in \mathbb{R}^{3 \times 3}$ such that*

$$\mathbf{X} = \begin{bmatrix} 0 & 0 & a \\ 0 & 0 & b \\ a & b & 0 \end{bmatrix}, \tag{10}$$

*the singular value decomposition of $\mathbf{X}$ is given as $\mathbf{X} = \mathbf{U}\mathbf{S}\mathbf{W}^T$, where*

$$\mathbf{U} = \begin{bmatrix} \frac{a}{\sqrt{a^2+b^2}} & 0 & \frac{-b}{\sqrt{a^2+b^2}} \\ \frac{b}{\sqrt{a^2+b^2}} & 0 & \frac{a}{\sqrt{a^2+b^2}} \\ 0 & -1 & 0 \end{bmatrix}, \ \mathbf{S} = \begin{bmatrix} \sqrt{a^2+b^2} & 0 & 0 \\ 0 & \sqrt{a^2+b^2} & 0 \\ 0 & 0 & 0 \end{bmatrix} \ and$$

$$\mathbf{W} = \begin{bmatrix} 0 & \frac{-a}{\sqrt{a^2+b^2}} & \frac{-b}{\sqrt{a^2+b^2}} \\ 0 & \frac{-b}{\sqrt{a^2+b^2}} & \frac{a}{\sqrt{a^2+b^2}} \\ 1 & 0 & 0 \end{bmatrix}.$$

Now, we prove the theorems stated in the main paper.

37th Conference on Neural Information Processing Systems (NeurIPS 2023).

**Theorem 1.** *For a set of consistent directions, $\mathbf{v}_{ij}, (i,j) \in \mathcal{E}_\Delta$, in $\mathcal{G}_\Delta$, the absolute change in any eigenvalue of $\mathbf{V^T V}$, denoted as $|\delta\lambda|$, when the directions $\mathbf{v}_{ij}$ are perturbed by small rotations $\delta\mathbf{R}_{ij}$, with $\mathbf{n}_{ij}$ and $\delta\theta_{ij} > 0$ being the rotation axis and angle, is bounded by*

$$|\delta\lambda| \leq \sum_{(k,i,j)\in TI(\Delta)} \delta\theta_{ij} \cdot \|\mathbf{v}_{ki}^{\mathbf{n}_{ij}\perp}\| \cdot \|\mathbf{v}_{kj}^{\mathbf{n}_{ij}\perp}\| \cdot \frac{\left|\sin\phi_{(k,i),(k,j)}^{\mathbf{n}_{ij}\perp}\right|}{\sin^2\phi_{(k,i),(k,j)}} \cdot$$

$$\left[\left(1 + (\mathbf{v}_{ik}^T\mathbf{v}_{jk})^2\right)\left((\mathbf{v}_{ij}^T\mathbf{v}_{ik})^2 + (\mathbf{v}_{ij}^T\mathbf{v}_{jk})^2\right) - 4 \cdot \mathbf{v}_{ik}^T\mathbf{v}_{jk} \cdot \mathbf{v}_{ij}^T\mathbf{v}_{ik} \cdot \mathbf{v}_{ij}^T\mathbf{v}_{jk}\right]^{\frac{1}{2}}, \quad (2)$$

*where $TI(\Delta) = \{(1,2,3),(2,3,1),(3,1,2)\}$, $\mathbf{v}^{\mathbf{n}_{ij}\perp}$ is the component of $\mathbf{v}$ orthogonal to $\mathbf{n}_{ij}$, $\phi_{(k,i),(k,j)}^{\mathbf{n}_{ij}\perp}$ is the angle between $\mathbf{v}_{ki}^{\mathbf{n}_{ij}\perp}$ and $\mathbf{v}_{kj}^{\mathbf{n}_{ij}\perp}$, and $\phi_{(k,i),(k,j)}$ is the angle between $\mathbf{v}_{ki}$ and $\mathbf{v}_{kj}$.*

*Proof.* Our goal is to check for the absolute change in the eigenvalue of $\mathbf{V^T V}$, denoted as $|\delta\lambda|$. $\mathbf{V^T V}$ is a symmetric matrix. Using Result 1 and Eqn. 9, finding the matrix-2 norm of the perturbation on $\mathbf{V^T V}$ will give us the bound on the change in its eigenvalue.

Let us first look at the case when one edge is perturbed and then extend it later to all edges. We note that $\mathcal{E}_\Delta = \{(1,2),(2,3)(3,1)\}$. Without loss of generality, we perturb $\mathbf{v}_{31}$ and denote the perturbed vector as $\mathbf{v}_{31}^P$. Since the perturbation is made by a small rotation $\delta\mathbf{R}_{31}$, we can write the perturbed vector as

$$\begin{aligned}\mathbf{v}_{31}^P &= \delta\mathbf{R}_{31}\mathbf{v}_{31} \\ &= (\mathbf{I} + \delta\theta_{31}\left[\mathbf{n}_{31}\right]_\times)\mathbf{v}_{31} \\ &= \mathbf{v}_{31} + \delta\boldsymbol{\omega}_{31} \times \mathbf{v}_{31}, \quad (11)\end{aligned}$$

where $\delta\boldsymbol{\omega}_{31} = \delta\theta_{31}\mathbf{n}_{31}$. From Eqn. 11, it can be seen that the perturbation can be treated as an additive perturbation $\mathbf{v}_{31}^P = \mathbf{v}_{31} + \delta\mathbf{v}_{31}$, where $\delta\mathbf{v}_{31} = \delta\boldsymbol{\omega}_{31} \times \mathbf{v}_{31}$. Let the perturbed matrix of the directions be $\mathbf{V}^P$ such that $\mathbf{V}^P = \mathbf{V} + \delta\mathbf{V}_{31}$, where $\delta\mathbf{V}_{31} = [\mathbf{0} \quad \mathbf{0} \quad \delta\mathbf{v}_{31}]$. Then, $(\mathbf{V}^P)^T\mathbf{V}^P$ can be written as

$$\begin{aligned}(\mathbf{V}^P)^T\mathbf{V}^P &= (\mathbf{V} + \delta\mathbf{V}_{31})^T\left(\mathbf{V} + \delta\mathbf{V}_{31}\right) \\ &\approx \mathbf{V}^T\mathbf{V} + \mathbf{V}^T\delta\mathbf{V}_{31} + \delta\mathbf{V}_{31}^T\mathbf{V} \text{ [ignoring second order terms]} \\ &= \mathbf{V}^T\mathbf{V} + \delta_{31}(\mathbf{V}^T\mathbf{V}), \quad (12)\end{aligned}$$

where $\delta_{31}(\mathbf{V}^T\mathbf{V}) = \mathbf{V}^T\delta\mathbf{V}_{31} + \delta\mathbf{V}_{31}^T\mathbf{V}$. After simplification of $\delta_{31}(\mathbf{V}^T\mathbf{V})$, we get

$$\delta_{31}(\mathbf{V}^T\mathbf{V}) = \begin{bmatrix} 0 & 0 & \delta\boldsymbol{\omega}_{31}^T(\mathbf{v}_{31} \times \mathbf{v}_{12}) \\ 0 & 0 & \delta\boldsymbol{\omega}_{31}^T(\mathbf{v}_{31} \times \mathbf{v}_{23}) \\ \delta\boldsymbol{\omega}_{31}^T(\mathbf{v}_{31} \times \mathbf{v}_{12}) & \delta\boldsymbol{\omega}_{31}^T(\mathbf{v}_{31} \times \mathbf{v}_{23}) & 0 \end{bmatrix}. \quad (13)$$

It can be seen that $\delta_{31}(\mathbf{V}^T\mathbf{V})$ has the same structure as that of the matrix in Eqn. 10. So, using Result 2, we get

$$\sigma_{max}(\delta_{31}(\mathbf{V}^T\mathbf{V})) = \left\|\begin{bmatrix}\delta\omega_{31}^T(\mathbf{v}_{31} \times \mathbf{v}_{12}) \\ \delta\omega_{31}^T(\mathbf{v}_{31} \times \mathbf{v}_{23})\end{bmatrix}\right\|. \quad (14)$$

Since the directions are consistent, the three directions are coplanar. So, there exists $\alpha$ and $\beta$ such that

$$\mathbf{v}_{31} = \alpha\mathbf{v}_{12} + \beta\mathbf{v}_{23}. \quad (15)$$

We obtain $\alpha$ and $\beta$ by premultiplying Eqn. 15 by $\mathbf{v}_{12}^T$ and $\mathbf{v}_{23}^T$ to get the following set of equations

$$\mathbf{v}_{12}^T\mathbf{v}_{31} = \alpha\mathbf{v}_{12}^T\mathbf{v}_{12} + \beta\mathbf{v}_{12}^T\mathbf{v}_{23}, \quad (16)$$

$$\mathbf{v}_{23}^T\mathbf{v}_{31} = \alpha\mathbf{v}_{23}^T\mathbf{v}_{12} + \beta\mathbf{v}_{23}^T\mathbf{v}_{23}. \quad (17)$$

Solving Eqns. 16 and 17 and using the fact that $\mathbf{v}_{ij}^T\mathbf{v}_{ij} = 1$, we get

$$\alpha = \frac{\mathbf{v}_{31}^T\mathbf{v}_{12} - (\mathbf{v}_{23}^T\mathbf{v}_{31})(\mathbf{v}_{23}^T\mathbf{v}_{12})}{1 - (\mathbf{v}_{12}^T\mathbf{v}_{23})^2}, \quad (18)$$

$$\beta = \frac{\mathbf{v}_{31}^T\mathbf{v}_{23} - (\mathbf{v}_{12}^T\mathbf{v}_{31})(\mathbf{v}_{12}^T\mathbf{v}_{23})}{1 - (\mathbf{v}_{12}^T\mathbf{v}_{23})^2}. \quad (19)$$

Also, using Eqn. 15, we get

$$\mathbf{v}_{31} \times \mathbf{v}_{12} = \beta(\mathbf{v}_{23} \times \mathbf{v}_{12}), \tag{20}$$

$$\mathbf{v}_{31} \times \mathbf{v}_{23} = \alpha(\mathbf{v}_{12} \times \mathbf{v}_{23}). \tag{21}$$

Using Eqns. 20 and 21 in Eqn. 14, we get

$$\sigma_{max}(\delta_{31}(\mathbf{V}^T\mathbf{V})) = \left\| \begin{bmatrix} \beta \cdot \delta\omega_{31}^T(\mathbf{v}_{23} \times \mathbf{v}_{12}) \\ \alpha \cdot \delta\omega_{31}^T(\mathbf{v}_{12} \times \mathbf{v}_{23}) \end{bmatrix} \right\| = \delta\theta_{31} \left\| \begin{bmatrix} \beta \cdot \mathbf{n}_{31}^T(\mathbf{v}_{23} \times \mathbf{v}_{12}) \\ \alpha \cdot \mathbf{n}_{31}^T(\mathbf{v}_{12} \times \mathbf{v}_{23}) \end{bmatrix} \right\|. \tag{22}$$

Let us decompose the directions as $\mathbf{v} = \mathbf{v}^{\mathbf{n}\|} + \mathbf{v}^{\mathbf{n}\perp}$, where $\mathbf{v}^{\mathbf{n}\|}$ and $\mathbf{v}^{\mathbf{n}\perp}$ denote parallel and perpendicular components of $\mathbf{v}$ to $\mathbf{n}$, which can be obtained as

$$\mathbf{v}^{\mathbf{n}\|} = (\mathbf{v}^T\mathbf{n}) \cdot \mathbf{n}, \tag{23}$$

$$\mathbf{v}^{\mathbf{n}\perp} = \mathbf{v} - \mathbf{v}^{\mathbf{n}\|} = \mathbf{v} - (\mathbf{v}^T\mathbf{n}) \cdot \mathbf{n}. \tag{24}$$

Using Eqns. 23 and 24, we get

$$\begin{aligned}
\mathbf{v}_{23} \times \mathbf{v}_{12} &= \left( (\mathbf{v}_{23}^T\mathbf{n}_{31}) \cdot \mathbf{n}_{31} + \mathbf{v}_{23}^{\mathbf{n}_{31}\perp} \right) \times \left( (\mathbf{v}_{12}^T\mathbf{n}_{31}) \cdot \mathbf{n}_{31} + \mathbf{v}_{12}^{\mathbf{n}_{31}\perp} \right) \\
&= \left( (\mathbf{v}_{23}^T\mathbf{n}_{31}) \cdot \mathbf{n}_{31} \right) \times \left( (\mathbf{v}_{12}^T\mathbf{n}_{31}) \cdot \mathbf{n}_{31} \right) + (\mathbf{v}_{23}^T\mathbf{n}_{31}) \cdot \mathbf{n}_{31} \times \mathbf{v}_{12}^{\mathbf{n}_{31}\perp} \\
&\quad + \mathbf{v}_{23}^{\mathbf{n}_{31}\perp} \times (\mathbf{v}_{12}^T\mathbf{n}_{31}) \cdot \mathbf{n}_{31} + \mathbf{v}_{23}^{\mathbf{n}_{31}\perp} \times \mathbf{v}_{12}^{\mathbf{n}_{31}\perp} \\
&= (\mathbf{v}_{23}^T\mathbf{n}_{31}) \cdot \left( \mathbf{n}_{31} \times \mathbf{v}_{12}^{\mathbf{n}_{31}\perp} \right) + (\mathbf{v}_{12}^T\mathbf{n}_{31}) \cdot \left( \mathbf{v}_{23}^{\mathbf{n}_{31}\perp} \times \mathbf{n}_{31} \right) + \left( \mathbf{v}_{23}^{\mathbf{n}_{31}\perp} \times \mathbf{v}_{12}^{\mathbf{n}_{31}\perp} \right).
\end{aligned} \tag{25}$$

Premultiplying $\mathbf{n}_{31}^T$ in Eqn. 25, we get

$$\mathbf{n}_{31}^T (\mathbf{v}_{23} \times \mathbf{v}_{12}) = \mathbf{n}_{31}^T \left( \mathbf{v}_{23}^{\mathbf{n}_{31}\perp} \times \mathbf{v}_{12}^{\mathbf{n}_{31}\perp} \right) = \|\mathbf{v}_{23}^{\mathbf{n}_{31}\perp} \times \mathbf{v}_{12}^{\mathbf{n}_{31}\perp}\|, \tag{26}$$

since, by construction, $\mathbf{v}_{23}^{\mathbf{n}_{31}\perp} \times \mathbf{v}_{12}^{\mathbf{n}_{31}\perp} = \|\mathbf{v}_{23}^{\mathbf{n}_{31}\perp} \times \mathbf{v}_{12}^{\mathbf{n}_{31}\perp}\| \cdot \mathbf{n}_{31}$. Using Eqn. 26 into Eqn. 22, we get

$$\sigma_{max}(\delta_{31}(\mathbf{V}^T\mathbf{V})) = \delta\theta_{31} \cdot \|\mathbf{v}_{12}^{\mathbf{n}_{31}\perp} \times \mathbf{v}_{23}^{\mathbf{n}_{31}\perp}\| \cdot \left\| \begin{bmatrix} -\beta \\ \alpha \end{bmatrix} \right\|. \tag{27}$$

Using the values of $\alpha$ and $\beta$ from Eqns. 18 and 19, we get

$$\begin{aligned}
\sigma_{max}(\delta_{31}(\mathbf{V}^T\mathbf{V})) &= \delta\theta_{31} \cdot \frac{\|\mathbf{v}_{12}^{\mathbf{n}_{31}\perp} \times \mathbf{v}_{23}^{\mathbf{n}_{31}\perp}\|}{1 - (\mathbf{v}_{12}^T\mathbf{v}_{23})^2} \cdot \\
&\quad [(1 + (\mathbf{v}_{12}^T\mathbf{v}_{23})^2) \cdot ((\mathbf{v}_{23}^T\mathbf{v}_{31})^2 + (\mathbf{v}_{12}^T\mathbf{v}_{31})^2) - 4 \cdot \mathbf{v}_{12}^T\mathbf{v}_{23} \cdot \mathbf{v}_{23}^T\mathbf{v}_{31} \cdot \mathbf{v}_{31}^T\mathbf{v}_{12}]^{\frac{1}{2}} \\
&= \delta\theta_{31} \cdot \|\mathbf{v}_{12}^{\mathbf{n}_{31}\perp}\| \cdot \|\mathbf{v}_{23}^{\mathbf{n}_{31}\perp}\| \cdot \frac{|\sin\phi_{(2,1),(2,3)}^{\mathbf{n}_{ij}\perp}|}{\sin\phi_{(2,1),(2,3)}^2} \cdot \\
&\quad [(1 + (\mathbf{v}_{12}^T\mathbf{v}_{23})^2) \cdot ((\mathbf{v}_{23}^T\mathbf{v}_{31})^2 + (\mathbf{v}_{12}^T\mathbf{v}_{31})^2) - 4 \cdot \mathbf{v}_{12}^T\mathbf{v}_{23} \cdot \mathbf{v}_{23}^T\mathbf{v}_{31} \cdot \mathbf{v}_{31}^T\mathbf{v}_{12}]^{\frac{1}{2}},
\end{aligned} \tag{28}$$

where $\phi_{(2,1),(2,3)}^{\mathbf{n}_{31}\perp}$ is the angle between $\mathbf{v}_{21}^{\mathbf{n}_{31}\perp}$ and $\mathbf{v}_{23}^{\mathbf{n}_{31}\perp}$, and $\phi_{(2,1),(2,3)}$ is the angle between $\mathbf{v}_{21}$ and $\mathbf{v}_{23}$.

Now, we perturb all three edges in $\mathcal{G}_\Delta$. Then, Eqn. 12 will change to

$$\begin{aligned}
(\mathbf{V}^P)^T\mathbf{V}^P &= (\mathbf{V} + \delta\mathbf{V}_{12} + \delta\mathbf{V}_{23} + \delta\mathbf{V}_{31})^T (\mathbf{V} + \delta\mathbf{V}_{12} + \delta\mathbf{V}_{23} + \delta\mathbf{V}_{31}) \\
&\approx \mathbf{V}^T\mathbf{V} + \mathbf{V}^T\delta\mathbf{V}_{12} + \delta\mathbf{V}_{12}^T\mathbf{V} + \mathbf{V}^T\delta\mathbf{V}_{23} + \delta\mathbf{V}_{23}^T\mathbf{V} + \mathbf{V}^T\delta\mathbf{V}_{31} + \delta\mathbf{V}_{31}^T\mathbf{V} \\
&\quad \text{[ignoring second order terms]} \\
&= \mathbf{V}^T\mathbf{V} + \delta_{12}(\mathbf{V}^T\mathbf{V}) + \delta_{23}(\mathbf{V}^T\mathbf{V}) + \delta_{31}(\mathbf{V}^T\mathbf{V}).
\end{aligned} \tag{29}$$

Using Result 1, we get,

$$\begin{aligned}
|\delta\lambda| &\leq \|\delta_{12}(\mathbf{V}^T\mathbf{V}) + \delta_{23}(\mathbf{V}^T\mathbf{V}) + \delta_{31}(\mathbf{V}^T\mathbf{V})\|_2 \\
&\leq \|\delta_{12}(\mathbf{V}^T\mathbf{V})\|_2 + \|\delta_{23}(\mathbf{V}^T\mathbf{V})\|_2 + \|\delta_{31}(\mathbf{V}^T\mathbf{V})\|_2 \text{ [using triangle inequality]}. \tag{30}
\end{aligned}$$

Following Eqn. 28 for each edge perturbation in Eqn. 30, we get

$$|\delta\lambda| \leq \sum_{(k,i,j)\in TI(\Delta)} \delta\theta_{ij} \cdot \|\mathbf{v}_{ki}^{\mathbf{n}_{ij}\perp}\| \cdot \|\mathbf{v}_{kj}^{\mathbf{n}_{ij}\perp}\| \cdot \frac{\left|\sin\phi_{(k,i),(k,j)}^{\mathbf{n}_{ij}\perp}\right|}{\sin^2\phi_{(k,i),(k,j)}} \cdot$$
$$\left[\left(1 + (\mathbf{v}_{ik}^T\mathbf{v}_{jk})^2\right)\left((\mathbf{v}_{ij}^T\mathbf{v}_{ik})^2 + (\mathbf{v}_{ij}^T\mathbf{v}_{jk})^2\right) - 4\cdot\mathbf{v}_{ik}^T\mathbf{v}_{jk}\cdot\mathbf{v}_{ij}^T\mathbf{v}_{ik}\cdot\mathbf{v}_{ij}^T\mathbf{v}_{jk}\right]^{\frac{1}{2}},$$

where $TI(\Delta) = \{(1,2,3),(2,3,1),(3,1,2)\}$. $\blacksquare$

**Corollary 1.** *For a set of consistent directions, $\mathbf{v}_{ij}, (i,j) \in \mathcal{E}_\Delta$, in $\mathcal{G}_\Delta$, the absolute change in any eigenvalue of $\mathbf{V^TV}$, denoted as $|\delta\lambda|$, when the directions $\mathbf{v}_{ij}$ are perturbed by small rotations $\delta\mathbf{R}_{ij}$, with $\mathbf{n}_{ij}$ and $\delta\theta_{ij} > 0$ being the rotation axis and angle, and $\mathbf{n}_{ij}$ being othogonal to $\mathbf{v}_{ki}$ and $\mathbf{v}_{kj}$ for all $(i,j,k) \in TI(\Delta)$, is bounded by*

$$|\delta\lambda| \leq \sum_{(k,i,j)\in TI(\Delta)} \delta\theta_{ij} \cdot \frac{1}{|\sin\phi_{(k,i),(k,j)}|} \cdot$$
$$\left[\left(1 + (\mathbf{v}_{ik}^T\mathbf{v}_{jk})^2\right)\left((\mathbf{v}_{ij}^T\mathbf{v}_{ik})^2 + (\mathbf{v}_{ij}^T\mathbf{v}_{jk})^2\right) - 4\cdot\mathbf{v}_{ik}^T\mathbf{v}_{jk}\cdot\mathbf{v}_{ij}^T\mathbf{v}_{ik}\cdot\mathbf{v}_{ij}^T\mathbf{v}_{jk}\right]^{\frac{1}{2}}, \quad (3)$$

*where $TI(\Delta) = \{(1,2,3),(2,3,1),(3,1,2)\}$, and $\phi_{(k,i),(k,j)}$ is the angle between $\mathbf{v}_{ki}$ and $\mathbf{v}_{kj}$.*

*Proof.* The unperturbed directions $\mathbf{v}_{ij}$ are coplanar since the directions are consistent. Given that the rotation axis $\mathbf{n}_{ij}$ for all the rotations $\delta\mathbf{R}_{ij}$ are orthogonal to $\mathbf{v}_{ki}$ and $\mathbf{v}_{kj}$ for all $(i,j,k) \in TI(\Delta)$, the rotation axis $\mathbf{n}_{ij}$ is orthogonal to $\mathbf{v}_{kl}$, for all $(i,j),(k,l) \in \mathcal{E}_\Delta$. This ensures that the perturbed directions are still coplanar and, thus, are consistent. Since $\mathbf{n}_{ij}$ is orthogonal to all the directions $\mathbf{v}_{ij}$, the component of directions parallel to the axis are zero, i.e. $\mathbf{v}^{\mathbf{n}\|} = \mathbf{0}$. So, $\mathbf{v}^{\mathbf{n}\perp} = \mathbf{v}$ (refer to Eqns. 23 and 24) and $\sin\phi_{(k,i),(k,j)}^{\mathbf{n}_{ij}\perp} = \sin\phi_{(k,i),(k,j)}$. Also, $\|\mathbf{v}_{ij}^{\mathbf{n}\perp}\| = \|\mathbf{v}_{ij}\| = 1$ for all $(i,j) \in \mathcal{E}_\Delta$. Hence, using these in Eqn. 2, we get

$$|\delta\lambda| \leq \sum_{(k,i,j)\in TI(\Delta)} \delta\theta_{ij} \cdot \frac{1}{|\sin\phi_{(k,i),(k,j)}|} \cdot$$
$$\left[\left(1 + (\mathbf{v}_{ik}^T\mathbf{v}_{jk})^2\right)\left((\mathbf{v}_{ij}^T\mathbf{v}_{ik})^2 + (\mathbf{v}_{ij}^T\mathbf{v}_{jk})^2\right) - 4\cdot\mathbf{v}_{ik}^T\mathbf{v}_{jk}\cdot\mathbf{v}_{ij}^T\mathbf{v}_{ik}\cdot\mathbf{v}_{ij}^T\mathbf{v}_{jk}\right]^{\frac{1}{2}}.$$

$\blacksquare$

**Theorem 2.** *Consider the bearing based-network of 3 nodes and 3 edges, $\mathcal{G}_\Delta = (\mathcal{V}_\Delta, \mathcal{E}_\Delta)$, and the corresponding angle matrix $\mathbf{A}_{\mathcal{G}_\Delta}$. The conditioning of the matrix $\mathbf{A}_{\mathcal{G}_\Delta}$ signifies the skewness of the triangle formed using the directions in $\mathcal{E}_\Delta$.*

*Proof.* The determinant of $\mathbf{A}_{\mathcal{G}_\Delta}$ is given as $|\mathbf{A}_{\mathcal{G}_\Delta}| = 2 \cdot \phi_{(1,2),(1,3)} \cdot \phi_{(2,1),(2,3)} \cdot \phi_{(3,1),(3,2)}$. The determinant will be zero if and only if atleast one of the angles in the triangle is zero. So, the matrix is non-singular when all the angles are non-zero.

It is known that the determinant is not a good measure of the closeness of the matrix to singularity [39]. So, we consider the closeness of any two columns in the matrix in terms of the cosine of the angles between the column vectors of $\mathbf{A}_{\mathcal{G}_\Delta}$. Let $\mathbf{a}_{\mathcal{G}_\Delta}^{ij}$ be the column of $\mathbf{A}_{\mathcal{G}_\Delta}$ corresponding to the edge $(i,j) \in \mathcal{E}_\Delta$. In $\mathcal{G}_\Delta$, there is one node common for any pair of edges. So, the cosine of the angle between any two columns is given as $\cos\angle(\mathbf{a}_{\mathcal{G}_\Delta}^{ij}, \mathbf{a}_{\mathcal{G}_\Delta}^{jk}) = \frac{\phi_{(i,j),(i,k)}\cdot\phi_{(k,i),(k,j)}}{\sqrt{\phi_{(i,j),(i,k)}^2+\phi_{(j,i),(j,k)}^2}\cdot\sqrt{\phi_{(j,i),(j,k)}^2+\phi_{(k,i),(k,j)}^2}}$.
It can be clearly seen that if the angle at the common node $j$ is small, i.e. $\phi_{(j,i),(j,k)} \to 0$ radian, then the two columns are very close or $\cos\angle(\mathbf{a}_{\mathcal{G}_\Delta}^{ij}, \mathbf{a}_{\mathcal{G}_\Delta}^{jk}) \to 1$. Small values of $\phi_{(j,i),(j,k)}$ make $\mathbf{A}_{\mathcal{G}_\Delta}$ nearly singular and its condition number high. If all values of $\phi_{(j,i),(j,k)}$ are sufficiently large, then $\cos\angle(\mathbf{a}_{\mathcal{G}_\Delta}^{ij}, \mathbf{a}_{\mathcal{G}_\Delta}^{jk})$ between any two columns will not be close to 1 making $\mathbf{A}_{\mathcal{G}_\Delta}$ well conditioned with a low condition number. Thus, the condition number of $\mathbf{A}_{\mathcal{G}_\Delta}$ is reflective of the skewness of the triangle formed from the directions in $\mathcal{E}_\Delta$. $\blacksquare$

**Theorem 3.** *Consider a bearing-based network $\mathcal{G}$, with all edges contributing to triplets. The angle matrix $\mathbf{A}_{\mathcal{G}}$, corresponding to $\mathcal{G}$, is well conditioned if the minimum angle (or equivalently all the angles) in all the triangles formed by the triplets are sufficiently large.*

*Proof.* Let us assume that all the angles in all the triangles formed by the triplets in $\mathcal{G}$ are sufficiently large. In $\mathbf{A}_{\mathcal{G}}$, no two columns can have exactly the same set of non-zero elements. This is evident since the diagonal entries are zero. In real-world scenarios, $\mathcal{G}$ is a sparse network, and thus $\mathbf{A}_{\mathcal{G}}$ is sparse. If two columns have exactly the same set of non-zero elements, then it means the two edges share exactly the same set of triplets. This is attained only when the two edges are connecting the same two nodes. But this is not possible since multiple edges between nodes are not allowed. Also, every column will have atleast two non-zero elements since every edge is a part of atleast one triplet and angles measured with the common nodes between edges will be present. This makes the angle matrix non-singular when all the angles in all the triangles are sufficiently large.

Next, we look at the conditioning of the matrix. Let $\mathbf{a}_{\mathcal{G}}^{ij}$ be the column of $\mathbf{A}_{\mathcal{G}}$ corresponding to the edge $(i,j) \in \mathcal{E}$. We check the cosine of the angle between any two columns of $\mathbf{A}_{\mathcal{G}}$ to understand the conditioning of $\mathbf{A}_{\mathcal{G}}$. We define triplets to be *adjacent* if they share a common edge, otherwise they are *non-adjacent*. There are only three cases which arise between any two edges $(i,j)$ and $(k,l)$, $(i,j),(k,l) \in \mathcal{E}$, and are listed below:

1. $(i,j)$ and $(k,l)$ belong to the same triplet,

2. $(i,j)$ and $(k,l)$ belong to adjacent triplets,

3. $(i,j)$ and $(k,l)$ belong to non-adjacent triplets.

Since each column in $\mathbf{A}_{\mathcal{G}}$ represents an edge, there are only the three cases, stated above, which occur for the pair of columns. For all three cases, the cosine of angles between two columns is checked. Let $Trp$ be the set of all triplets in $\mathcal{G}$.

**Case 1:** Edges $(i,j)$ and $(k,l)$ belong to the same triplet.

$$\cos \angle(\mathbf{a}_{\mathcal{G}}^{ij}, \mathbf{a}_{\mathcal{G}}^{kl})$$

$$= \frac{\phi_{(c_1',c),(c_1',c_2')} \cdot \phi_{(c_2',c_1'),(c_2',c)}}{\sqrt{\sum_{p:(i,j,p) \in Trp} \left( \phi_{(i,j),(i,p)}^2 + \phi_{(j,i),(j,p)}^2 \right)} \cdot \sqrt{\sum_{q:(k,l,q) \in Trp} \left( \phi_{(k,l),(k,p)}^2 + \phi_{(l,k),(l,q)}^2 \right)}},$$

where $c \in \{i,j\} \cap \{k,l\}$ is the common node and $c_1', c_2' \in \{i,j\} \cup \{k,l\} \setminus \{c\}$ with $c_1' \neq c_2'$ are the other two nodes in the triplet.

It can be seen that for this case, if all the angles are sufficiently large, then $\cos \angle(\mathbf{a}_{\mathcal{G}}^{ij}, \mathbf{a}_{\mathcal{G}}^{kl})$ cannot be close to 1, and thus the two columns are not similar.

**Case 2:** Edges $(i,j)$ and $(k,l)$ belong to adjacent triplets $tr1, tr2 \in Trp$.

*Part I:* When the edges $(i,j)$ and $(k,l)$ share a common node of the triplets $tr1$ and $tr2$:

$$\cos \angle(\mathbf{a}_{\mathcal{G}}^{ij}, \mathbf{a}_{\mathcal{G}}^{kl})$$

$$= \frac{\phi_{(r,c),(r,c_1')} \cdot \phi_{(r,c),(r,c_2')}}{\sqrt{\sum_{p:(i,j,p) \in Trp} \left( \phi_{(i,j),(i,p)}^2 + \phi_{(j,i),(j,p)}^2 \right)} \cdot \sqrt{\sum_{q:(k,l,q) \in Trp} \left( \phi_{(k,l),(k,p)}^2 + \phi_{(l,k),(l,q)}^2 \right)}},$$

where $c \in \{i,j\} \cap \{k,l\}$ is the common node, $(r,c) \in \mathcal{E}$ is the common edge between $tr1$ and $tr2$, and $c_1' \in \{i,j\} \setminus \{c\}, c_2' \in \{k,l\} \setminus \{c\}$ are the other two nodes in the two triplets.

*Part II:* When the edges $(i,j)$ and $(k,l)$ are connected by the common edge of the triplets $tr1$ and $tr2$:

$$\cos \angle(\mathbf{a}_{\mathcal{G}}^{ij}, \mathbf{a}_{\mathcal{G}}^{kl})$$

$$= \frac{\phi_{(c_1',c_1''),(c_1',c_2')} \cdot \phi_{(c_2',c_1'),(c_2',c_2'')}}{\sqrt{\sum_{p:(i,j,p) \in Trp} \left( \phi_{(i,j),(i,p)}^2 + \phi_{(j,i),(j,p)}^2 \right)} \cdot \sqrt{\sum_{q:(k,l,q) \in Trp} \left( \phi_{(k,l),(k,p)}^2 + \phi_{(l,k),(l,q)}^2 \right)}},$$

where $(c'_1, c'_2) \in \mathcal{E}$ is the common edge between $tr1$ and $tr2$ such that $c'_1, c'_2 \in \{i, j, k, l\}$ with $c'_1 \neq c'_2$, and $c''_1, c''_2 \in \{i, j, k, l\} \setminus \{c'_1, c'_2\}$ with $c''_1 \neq c''_2$ are the other two nodes in the triplet such that $(c'_1, c'_2, c''_1), (c'_1, c'_2, c''_2) \in Trp$.

It can be seen that for both parts in this case, if all the angles are sufficiently large, then $\cos \angle(\mathbf{a}_{\mathcal{G}}^{ij}, \mathbf{a}_{\mathcal{G}}^{kl})$ cannot be close to 1, and thus the two columns are not similar.

**Case 3:** Edges $(i, j)$ and $(k, l)$ belong to non-adjacent triplets $tr1, tr2 \in Trp$.

By the construction of the matrix using Eqn. 5, there would be no common non-zero elements in the two columns. So,

$$\cos \angle(\mathbf{a}_{\mathcal{G}}^{ij}, \mathbf{a}_{\mathcal{G}}^{kl}) = 0.$$

Thus, if all the angles in all the triangles formed by the triplets in $\mathcal{G}$ are sufficiently large, then the angle matrix $\mathbf{A}_{\mathcal{G}}$ is non-singular, and the columns are not similar implying that $\mathbf{A}_{\mathcal{G}}$ is well-conditioned. ∎

**Theorem 4.** *Given a bearing-based network $\mathcal{G}$, with all edges contributing to triplets forming triangles, and its corresponding triplet network $\mathcal{G}_T$, the maximal parallel rigid component of $\mathcal{G}$ can be determined by the edges in $\mathcal{G}$ contributing to the largest connected component of $\mathcal{G}_T$.*

*Proof.* It is known that the union of parallel rigid components is parallel rigid if the components share atleast one edge [2]. Each triplet forming a triangle in $\mathcal{G}$ is parallel rigid by itself. Also, by construction, each connected component of the triplet network $\mathcal{G}_T$ will have shared edges among triplets forming triangles, making every component parallel rigid independently. Thus, the maximal parallel rigid graph of $\mathcal{G}$ can be determined by the edges in $\mathcal{G}$ participating in the maximum connected component of $\mathcal{G}_T$. ∎

# 9   Proposed Method for Dense Networks

In Sec. 5 of the main paper, we presented the algorithm to remove skewed triangles from a sparse network $\mathcal{G}$. Here, we consider the case when $\mathcal{G}$ is dense (nearly all possible edges exist), making $\mathbf{A}_{\mathcal{G}}$ dense. We provide the method for removing skewed triangles in the following steps.

**Step 1:** First, construct a matrix $\mathbf{V}$ with columns as the directions in $\mathcal{G}$. The elements of $\mathbf{V}^{\mathbf{T}}\mathbf{V}$ give the dot product between all possible directions. Then, construct the boundary matrix using Eqn. 6. The row and column indices of non-zero elements of the matrix $\mathbf{B}_2 \mathbf{B}_2^T$ give us the edge pairs participating in triplets which are the non-zero elements in the angle matrix $\mathbf{A}_{\mathcal{G}}$. So, the zero elements of the matrix $\mathbf{B}_2 \mathbf{B}_2^T$ can be used to mask the elements in $\mathbf{V}^{\mathbf{T}}\mathbf{V}$. The net matrix generated is $\tilde{\mathbf{C}}_{\mathcal{G}}$.

**Step 2:** We identify the correct signs for the elements in $\tilde{\mathbf{C}}_{\mathcal{G}}$ to ensure that the dot products reflect the cosine of the angles of the triangles. We first create a matrix $\mathbf{M}$ consisting of $m^{ij,kl}$ (Eqn. 7) as the elements corresponding to the row for $(i, j)^{th}$ edge and column for $(k, l)^{th}$ edge. Then, the Hadamard product (element-wise multiplication) of $\tilde{\mathbf{C}}_{\mathcal{G}}$ and $\mathbf{M}$ gives $\mathbf{C}_{\mathcal{G}}$. Once the matrix $\mathbf{C}_{\mathcal{G}}$ is formed, we take element-wise inverse cosine to get the angles as the entries and thus the angle matrix $\mathbf{A}_{\mathcal{G}}$.

**Step 3:** This step is the same as that provided in the main paper.

We summarize the method in Algo. 2.

# 10   Formulations for Solving Translation Averaging

In Sec. 6 of the main paper, we used two formulations, Revised LUD [41, 57] and BATA [57], for the experiments. Here, we provide the optimization problems of those methods for ready reference.

**Algorithm 2:** Removal of Skewed Triangles from Dense Networks

---

**Input:** Bearing-based network $\mathcal{G} = (\mathcal{V}, \mathcal{E})$, containing only triplets and is parallel rigid, and the triplet list of $\mathcal{G}$.

**Output:** Bearing-based network $\mathcal{G}_F = (\mathcal{V}_F, \mathcal{E}_F)$, containing only triplets and is parallel rigid, without skewed triangles.

---

1 Compute the non-zero elements in angle matrix $\mathbf{A}_{\mathcal{G}}$ using the boundary matrix $\mathbf{B}_2$ (Eqn. 6).
2 Construct the matrix $\mathbf{V}$, whose columns consists of directions $\mathbf{v}_{ij}, (i,j) \in \mathcal{E}$.
3 Get dot products between all possible directions using $\mathbf{V}^T\mathbf{V}$.
4 Use zero elements in $\mathbf{B}_2^T\mathbf{B}_2$ to mask elements in $\mathbf{V}^T\mathbf{V}$, and denote it as $\tilde{\mathbf{C}}_{\mathcal{G}}$.
5 Get the signs of the dot product for non-zero elements of $\tilde{\mathbf{C}}_{\mathcal{G}}$, i.e. $m^{ij,kl}$ using Eqn. 7.
6 Construct a matrix $\mathbf{M}$ using $m^{ij,kl}$.
7 Perform element-wise multiplication of $\tilde{\mathbf{C}}_{\mathcal{G}}$ with $\mathbf{M}$ to get $\mathbf{C}_{\mathcal{G}}$.
8 Get the inverse cosine for every element in $\mathbf{C}_{\mathcal{G}}$ to get $\mathbf{A}_{\mathcal{G}}$.
9 Extract the angles of the triplets from $\mathbf{A}_{\mathcal{G}}$.
10 Filter out the triplets with the minimum angle less than a threshold to get $\tilde{\mathcal{G}}_F = (\tilde{\mathcal{V}}_F, \tilde{\mathcal{E}}_F)$.
11 Construct the triplet network $\mathcal{G}_T$ using the filtered network $\tilde{\mathcal{G}}_F$.
12 Get the largest connected component of $\mathcal{G}_T$.
13 Get the edges of $\tilde{\mathcal{G}}_F$ contributing to the largest connected component of $\mathcal{G}_T$ to get $\mathcal{G}_F = (\mathcal{V}_F, \mathcal{E}_F)$, which is connected and is parallel rigid.

---

**Revised LUD** [41, 57] (compares relative displacements):

$$\min_{\mathbf{T}_{i,i \in \mathcal{V}}, \lambda_{ij,(i,j) \in \mathcal{E}}} \sum_{(i,j) \in \mathcal{E}} \|\mathbf{T}_j - \mathbf{T}_i - \lambda_{ij}\mathbf{v}_{ij}\|_2 \tag{31}$$

$$\text{s.t.} \sum_{i \in \mathcal{V}} \mathbf{T}_i = \mathbf{0}, \sum_{(i,j) \in \mathcal{E}} \langle \mathbf{T}_j - \mathbf{T}_i, \mathbf{v}_{ij} \rangle = 1, \lambda_{ij} \geq 0, \forall (i,j) \in \mathcal{E}$$

**BATA** [57] (compares relative directions):

$$\min_{\mathbf{T}_{i,i \in \mathcal{V}}, \gamma_{ij,(i,j) \in \mathcal{E}}} \sum_{(i,j) \in \mathcal{E}} \rho\left(\|(\mathbf{T}_j - \mathbf{T}_i)\gamma_{ij} - \mathbf{v}_{ij}\|_2\right) \tag{32}$$

$$\text{s.t.} \sum_{i \in \mathcal{V}} \mathbf{T}_i = \mathbf{0}, \sum_{(i,j) \in \mathcal{E}} \langle \mathbf{T}_j - \mathbf{T}_i, \mathbf{v}_{ij} \rangle = 1, \gamma_{ij} \geq 0, \forall (i,j) \in \mathcal{E}$$

The zero centroid and dot product constraints in Eqns. 31 and 32 fix the origin and the global scale ambiguities, respectively. $\rho$ denotes the Cauchy loss function. $\lambda_{ij}$ and $\gamma_{ij}$ are non-negative variables that are ideally equal to baseline and inverse baseline for the edge $(i,j)$, respectively.

## 11 Additional Results

As mentioned in the main paper, our filtering scheme uses only input directions and does not favour any specific cost function used for translation averaging. In Sec. 6 of the main paper, we presented the results using BATA [57]. In the following subsection, we present the results using Revised LUD [41, 57] and observe a similar trend as seen for the BATA results.

### 11.1 Outlier-Free Data

In Sec. 6.2 of the main paper, we provided experimental results on outlier-free data. Here, in Table 5, we provide the details of the 1DSfM [55] datasets without outliers. It can be seen that only a small fraction of the nodes and edges are removed due to the removal of skewed triangles. Also, it can be seen that for many datasets, the condition number of the angle matrix (with matrix-2 norm) decreases. For remaining datasets, it either increases or is similar for both the unfiltered and filtered networks. As mentioned in Sec. 5 of the main paper, we perform non-aggressive pruning, due to which some

Table 5: Details of networks before and after removing skewed triangles from 1DSfM [55] datasets without outliers. $\#N_{rem}, \#M_{rem}$: No. of nodes and edges removed, $\kappa_2(\mathbf{A}_{\mathcal{G}})$: condition number of the angle matrix with matrix-2 norm, $t_{filter}$: time taken to remove skewed triangles.

| Dataset | #Nodes | | #Edges | | $\#N_{rem}$ | $\#M_{rem}$ | $\kappa_2(\mathbf{A}_{\mathcal{G}})$ | | $t_{filter}$ (sec) |
|---|---|---|---|---|---|---|---|---|---|
| | w/o filter | w/ filter | w/o filter | w/ filter | | | w/o filter | w/ filter | |
| ALM | 560 | 543 | 7351 | 7140 | 17 | 211 | 1.0e+12 | 1.0e+12 | 0.06 |
| ELS | 129 | 128 | 664 | 649 | 1 | 15 | 5.5e+05 | 2.4e+05 | 0.01 |
| GMM | 324 | 315 | 5231 | 5164 | 9 | 67 | 4.8e+07 | 2.5e+08 | 0.07 |
| MDR | 268 | 248 | 2545 | 2361 | 20 | 184 | 5.6e+07 | 2.6e+06 | 0.03 |
| MND | 445 | 439 | 10338 | 10175 | 6 | 163 | 7.8e+09 | 3.5e+09 | 0.25 |
| NYC | 337 | 317 | 3250 | 3065 | 20 | 185 | 9.9e+11 | 9.5e+11 | 0.02 |
| ND | 1199 | 1182 | 26414 | 25878 | 17 | 536 | 1.0e+12 | 1.0e+12 | 0.28 |
| PDP | 344 | 334 | 6196 | 5949 | 10 | 247 | 2.3e+08 | 3.3e+08 | 0.09 |
| PIC | 1844 | 1795 | 20880 | 20100 | 49 | 780 | 1.7e+11 | 1.2e+11 | 0.15 |
| ROF | 994 | 964 | 8696 | 8305 | 30 | 391 | 2.6e+10 | 2.6e+10 | 0.07 |
| TOL | 450 | 439 | 5017 | 4921 | 11 | 96 | 1.1e+09 | 5.6e+07 | 0.04 |
| TFG | 3962 | 3778 | 51450 | 49571 | 184 | 1879 | 1.1e+12 | 4.5e+09 | 0.52 |
| USQ | 403 | 382 | 3921 | 3682 | 21 | 239 | 3.8e+11 | 3.3e+11 | 0.04 |
| VNC | 724 | 705 | 12176 | 11901 | 19 | 275 | 2.3e+09 | 3.1e+09 | 0.17 |
| YKM | 385 | 372 | 4073 | 3892 | 13 | 181 | 3.1e+08 | 1.5e+09 | 0.04 |

Table 6: Absolute translation errors (in meters) on 1DSfM [55] datasets without outliers using Revised LUD [41, 57]. Removed Node Errors: Errors of removed nodes in the unfiltered network (meters), $t_{RLUD}$: time taken by Revised LUD.

| Dataset | Mean-ATE | | RMS-ATE | | Removed Node Errors | | $t_{RLUD}$ (sec) |
|---|---|---|---|---|---|---|---|
| | w/o filter | w/ filter | w/o filter | w/ filter | Mean | RMS | |
| ALM | 2.5 | **2.4** | 4.0 | **3.5** | 9.3 | 10.9 | 6 |
| ELS | **1.4** | 1.5 | **1.8** | 2.0 | 1.5 | 1.5 | 1 |
| GMM | 5.8 | **5.3** | 9.2 | **8.7** | 20.6 | 24.0 | 4 |
| MDR | 8.8 | **7.9** | 18.1 | **16.2** | 26.9 | 35.9 | 2 |
| MND | 2.3 | **2.2** | 3.2 | **3.2** | 4.7 | 5.7 | 7 |
| NYC | 2.7 | **2.6** | 5.6 | **5.4** | 5.6 | 7.4 | 3 |
| ND | **3.5** | **3.5** | 5.7 | **5.5** | 6.7 | 8.8 | 28 |
| PDP | 2.6 | **2.4** | 3.9 | **3.5** | 6.3 | 8.1 | 5 |
| PIC | 2.9 | **2.8** | 6.2 | **6.1** | 6.6 | 8.2 | 26 |
| ROF | **16.5** | 17.3 | **29.6** | 30.5 | 43.4 | 59.4 | 7 |
| TOL | 11.1 | **10.9** | 23.3 | **23.2** | 42.5 | 60.1 | 4 |
| TFG | 7.9 | **7.2** | 17.1 | **16.3** | 21.8 | 27.3 | 63 |
| USQ | 5.2 | **5.0** | 7.0 | **6.8** | 10.7 | 13.0 | 3 |
| VNC | 7.6 | **7.5** | 11.8 | **10.8** | 13.8 | 18.9 | 11 |
| YKM | **8.2** | 8.3 | 21.4 | **21.1** | 25.9 | 44.6 | 3 |

skewed triangles are still present. We re-emphasize that non-aggressive pruning still ensures that the nodes are estimated reliably since the nodes are also a part of atleast one triplet, which belongs to the set of non-skewed triangles.

In Table 6, we provide the results obtained from Revised LUD. It can be seen that the absolute translation errors decrease for most of the datasets after the removal of skewed triangles. Moreover, errors of the removed nodes in the unfiltered network are high compared to the overall errors, indicating that the removed nodes are not estimated reliably in the unfiltered network. We also observe that the time taken to filter skewed triangles (from Table 5) is $\sim 1\%$ of the time taken for translation averaging by Revised LUD (Table 6), which shows the efficiency of Algo. 1 of the main paper.

## 11.2 Real Data

In Sec. 6.3 of the main paper, we provided the details of the datasets and the translation averaging results using BATA [57]. In Table 7, we present the results using Revised LUD. It can be seen that similar to the trend in BATA results, the absolute translation estimates improve for most datasets. Also, the errors of the nodes removed due to filtering are high compared to the overall errors in the unfiltered network, indicating that the filtered nodes are not estimated reliably in the unfiltered network.

Table 7: Absolute translations errors (in meters) on 1DSfM [55] datasets using Revised LUD [41, 57]. Removed Node Errors: Errors of removed nodes in the unfiltered network (meters), $t_{RLUD}$: time taken by Revised LUD.

| Dataset | Mean-ATE | | RMS-ATE | | Removed Node Errors | | $t_{RLUD}$ (sec) |
|---------|----------|---------|----------|---------|------|------|------------------|
| | w/o filter | w/ filter | w/o filter | w/ filter | Mean | RMS | |
| ALM | 4.4 | **4.3** | 8.0 | **7.6** | 10.5 | 15.3 | 12 |
| ELS | 20.3 | **19.6** | **37.5** | 42.1 | 82.6 | 97.9 | 5 |
| GMM | **41.9** | 45.5 | **60.6** | 66.6 | 69.3 | 86.0 | 10 |
| MDR | 15.6 | **14.5** | 29.5 | **27.5** | 39.8 | 43.2 | 3 |
| MND | 3.9 | **3.8** | **5.8** | **5.8** | 12.7 | 13.4 | 12 |
| NYC | 4.5 | **4.2** | 8.2 | **7.3** | 18.9 | 24.9 | 5 |
| ND | 4.0 | **3.9** | **6.5** | **6.5** | 10.1 | 13.0 | 68 |
| PDP | 7.8 | **7.7** | 11.6 | **11.5** | 16.5 | 18.5 | 11 |
| PIC | 5.0 | **4.9** | **9.2** | 9.3 | 14.4 | 18.0 | 63 |
| ROF | 17.5 | **15.7** | 32.5 | **28.7** | 45.7 | 58.0 | 17 |
| TOL | 18.5 | **17.4** | 36.7 | **33.7** | 69.3 | 100.9 | 6 |
| TFG | 19.6 | **13.1** | 68.5 | **25.5** | 58.8 | 137.2 | 214 |
| USQ | 12.9 | **12.3** | 20.0 | **19.3** | 32.8 | 42.3 | 9 |
| VNC | **9.6** | 10.2 | **14.0** | 15.3 | 15.4 | 16.8 | 19 |
| YKM | 20.6 | **20.0** | 30.0 | **29.0** | 42.1 | 52.9 | 7 |

Table 8: Details of networks after applying 1DSfM outlier filter [55] without and with removal of skewed triangles (Algo. 1). $\#N_{rem}, \#M_{rem}$: No. of nodes and edges removed.

| Dataset | $\#N_{rem}$ | | $\#M_{rem}$ | |
|---------|-------------|---------|-------------|---------|
| Our filter → | w/o filter | w/ filter | w/o filter | w/ filter |
| ALM | 19 | 36 | 269 | 416 |
| ELS | 2 | 13 | 368 | 427 |
| GMM | 4 | 76 | 1062 | 1470 |
| MDR | 1 | 32 | 136 | 326 |
| MND | 3 | 7 | 1242 | 1312 |
| NYC | 2 | 17 | 251 | 443 |
| ND | 0 | 6 | 3790 | 3832 |
| PDP | 3 | 32 | 504 | 827 |
| PIC | 13 | 101 | 5229 | 6443 |
| ROF | 17 | 75 | 1953 | 2429 |
| TOL | 1 | 21 | 339 | 449 |
| USQ | 130 | 166 | 1769 | 1991 |
| VNC | 2 | 2 | 2120 | 2120 |
| YKM | 313 | 389 | 3223 | 3872 |

Next, we apply 1DSfM outlier filter (Algo. 2, [55]) on the datasets to study the complementary benefits of removal of skewed triangles after applying an outlier filter. We reiterate that the removal of skewed triangles and outlier edges are two distinct aspects of the problem. In Table 8, we provide the network details after applying 1DSfM outlier filter without and with the removal of skewed triangles and compare their performance in Table 9. It can be seen that applying the 1DSfM outlier filter and then removing skewed triangles with our filter leads to better translation estimates. Moreover, our filter + 1DSfM outlier filter removes the nodes which are poorly estimated than only 1DSfM outlier filter, as seen from removed node errors in the unfiltered network in Table 9. Since both outliers and skewed triangles are different issues, combining filters for both types improves accuracy.

In Sec. 5, we mentioned the improved speed using Algo. 1 compared to the brute force method. In Table 10, we compare the time taken by the brute force method to remove skewed triangles to that of our method in Algo. 1. In the brute force method, all three angles in every triangle are computed, and then the skewed triangles are identified and removed. We used 20 threads in parallel for the brute force method, while our method is based only on vectorized operations without any parallelization. It can be seen that Algo. 1 is significantly faster than the brute force method with $\sim 100$ times faster for most datasets and $\sim 1000$ times faster for large scale datasets. This shows the efficiency of Algo. 1.

Finally, we show the differences in reconstructions obtained without and with using our filter. In Fig. 4, the arch in the Notre Dame reconstruction is misplaced due to improper translation estimates, as seen in the lateral view. Using our filter removes such a misplaced arch. In Fig. 5, the wall in the

Table 9: Absolute translations errors (in meters) after applying 1DSfM [55] outlier filter without and with removal of skewed triangles (Algo. 1). Removed Node Errors: Errors of removed nodes in the unfiltered network.

| Dataset | Mean ATE | | RMS ATE | | Removed Nodes Errors (Mean) | | Removed Nodes Errors (RMS) | |
|---|---|---|---|---|---|---|---|---|
| Our filter → | w/o filter | w/ filter | w/o filter | w/ filter | w/o filter | w/ filter | w/o filter | w/ filter |
| ALM | 4.6 | **4.4** | **10.5** | **10.5** | 12.4 | 11.8 | 25.2 | 23.7 |
| ELS | 23.3 | **21.5** | 51.2 | **50.5** | 15.9 | 70.6 | 15.9 | 96.8 |
| GMM | 42.4 | **41.1** | 64.2 | **60.6** | 20.7 | 121.1 | 33.3 | 158.1 |
| MDR | 13.6 | **12.9** | 28.2 | **26.0** | 20.2 | 48.5 | 20.2 | 57.1 |
| MND | 4.5 | **4.3** | 10.0 | **9.8** | 9.5 | 20.9 | 15.3 | 26.5 |
| NYC | 6.6 | **5.2** | 15.4 | **12.1** | 9.4 | 36.8 | 9.4 | 46.4 |
| ND | **3.3** | **3.3** | **6.4** | **6.4** | 0 | 9.0 | 0 | 9.3 |
| PDP | **7.9** | 8.0 | **13.0** | 13.1 | 10.1 | 12.6 | 11.6 | 14.9 |
| PIC | 5.4 | **5.2** | 11.1 | **10.8** | 8.3 | 17.7 | 13.3 | 25.6 |
| ROF | **14.3** | 15.3 | 27.7 | **26.8** | 57.8 | 57.5 | 67.9 | 83.9 |
| TOL | 15.6 | **14.3** | 32.4 | **29.7** | 7.3 | 54.9 | 7.3 | 81.2 |
| USQ | **9.7** | 10.9 | **14.5** | 18.2 | 23.6 | 23.8 | 37.3 | 36.2 |
| VNC | 10.8 | **10.5** | **18.1** | 18.4 | 13.5 | 19.2 | 14.6 | 25.1 |
| YKM | 13.9 | **11.2** | 26.6 | **22.7** | 34.5 | 34.8 | 38.2 | 39.1 |

Table 10: Time taken (in sec) for removal of skewed triangles with brute force method and Algo. 1.

| Dataset | Brute Force | Algo. 1 |
|---|---|---|
| ALM | 20.55 | 0.31 |
| ELS | 5.28 | 0.14 |
| GMM | 12.40 | 0.20 |
| MDR | 0.96 | 0.05 |
| MND | 55.18 | 0.67 |
| NYC | 2.16 | 0.08 |
| ND | 1047.90 | 2.66 |
| PDP | 16.07 | 0.32 |
| PIC | 118.46 | 0.55 |
| ROF | 17.87 | 0.25 |
| TOL | 4.27 | 0.14 |
| TFG | 1110.40 | 1.83 |
| USQ | 11.33 | 0.23 |
| VNC | 72.63 | 0.65 |
| YKM | 7.36 | 0.18 |

Piazza del Popolo reconstruction is misplaced, but such an effect is not seen after applying our filter. This reveals that removing skewed triangles helps get more accurate reconstructions.

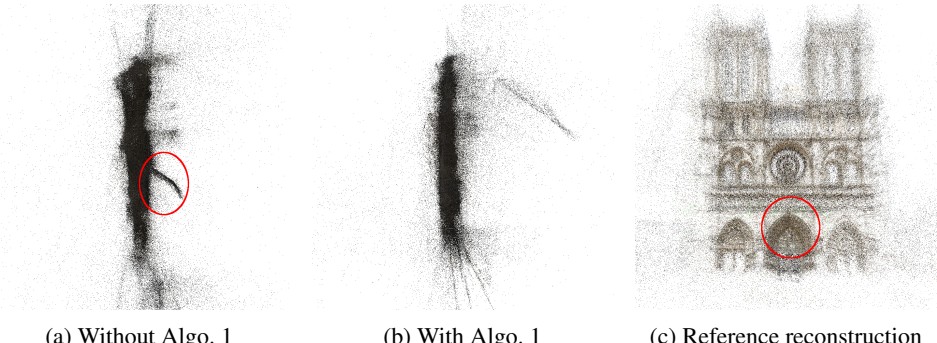

(a) Without Algo. 1       (b) With Algo. 1       (c) Reference reconstruction

Figure 4: Lateral view of Notre Dame [55] reconstruction. Improper reconstruction of the arch (red ellipse) obtained due to incorrect estimation of camera translation, which is not present after filtering skewed triangles with our method (Algo. 1). Reference reconstruction shows the position of the arch.

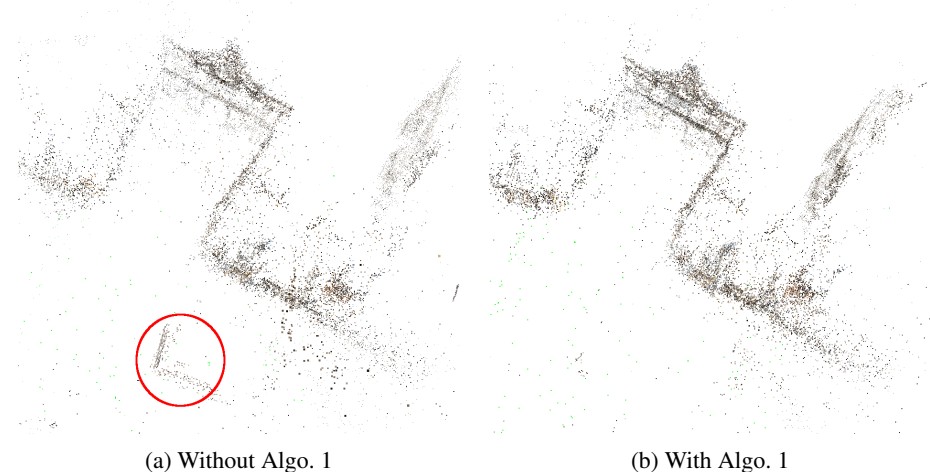

(a) Without Algo. 1       (b) With Algo. 1

Figure 5: Plan view of Piazza del Popolo [55] reconstruction. A misplaced wall (red ellipse) obtained due to incorrect estimation of camera translations, which is not present after filtering skewed triangles with our method (Algo. 1).

