# OpenReview forum: "Sensitivity in Translation Averaging"
_NeurIPS.cc/2023/Conference — NeurIPS 2023 poster_

### Official Review · Reviewer_Jy8r · 2023-06-19

**Soundness:** 3 good
**Presentation:** 3 good
**Contribution:** 3 good
**Rating:** 5
**Confidence:** 4

**Summary:**

The paper proposed a method to efficiently remove view triplets from a pose graph where the minimum angle falls below a threshold. This is relevant to global SfM algorithms, where such triplets (i.e., triangles) lead to highly uncertain translation scale estimation. The method's performance has been demonstrated on the 1DSfM dataset.

**Strengths:**

- The proposed method is a nice solution to a simple idea. Even though I feel the description has been written in an overcomplicated way, I like the idea and the method.
- The problem is relevant to global SfM methods where the translation averaging part is currently one of the main bottlenecks of achieving good accuracy.
- The proposed solution is ~100 times faster than the trivial brute-force solution.

**Weaknesses:**

I have two major problems with the proposed method, and both are regarding the provided experiments:
- First, the proposed method is not compared to any other filters (e.g., 1DSfM), which really makes it hard to understand how much it actually improves the SOTA (if it does). I know that the proposed solver can be easily combined with 1DSfM, but I am not convinced it leads to noticeable improvements in that case. To understand this, the authors should provide results of other filters with and without the proposed one.
- Second, the authors use the 1DSfM dataset in their experiments where the addressed problem of skewed triangles is not really present. This can be seen in Table 2, which shows that actually only a few edges are removed from the pose graphs (it is surprising that removing such few edges can lead to an improvement in the final global SfM accuracy). I think it would make more sense to showcase the method on datasets where SfM is actually often failing due to the coinciding translation directions, e.g., on KITTI, which presents the trajectory of a moving vehicle. There, scale estimation is challenging. On the 1DSfM dataset, most global methods work reasonably well. Making global SfM work on such cases would have an impact.

Without comparison to baselines, the paper is a clear rejection to me. In case such a comparison is provided, I am willing to improve my rating.

Minor things/comments:
- The brute-force solution's timing should be compared to the proposed method. I know that the authors mentioned this in the text, but it would be helpful to show the actual run-time, given that the speed-up is the main contribution of the paper, while accuracy-wise, the brute-force method should be similarly good.
- L109 "being the rotation axis angle" -> "being the rotation axis and angle respectively"
- Fig.1 How can the minimum angle in a triangle be close to 180°? Maybe I misunderstood the figure.
- L294 BATA is first mentioned here (aside from a half sentence in the introduction). It should be written down that BATA is used for estimating the global positions from the directions.
- Eq.2 is quite something that could be visualized by a figure and it would help a lot in understanding it.

**Questions:**

- I don't call this a major issue, but it is definitely something that should be discussed in the manuscript. It is unclear why the authors removed skewed triangles instead of deleting a single edge. The choice of triangle removal should be justified.

**Limitations:**

Discussed.

---

> ### Author Rebuttal · Authors · 2023-08-09
>
> 1. Results with 1DSfM filter: 1DSfM filter is designed to remove outlier edges while our method removes skewed triangles, which are two distinct aspects of the problem. We apply 1DSfM and then compare the solutions without and with applying our filter on real data. In Table R1 of the rebuttal pdf, we provide the comparison. It can be seen that applying both the 1DSfM filter and our filter leads to the overall best accuracy. Moreover, the improvements are significant for the datasets where the performance is better, while the accuracy is almost similar for the datasets where improvement is not seen. As noted earlier, since both outliers and skewed triangles are distinct issues, combining filters for both types leads to improved accuracy.
>
> 2. 1DSfM dataset for evaluation: In Fig. R2 of the rebuttal pdf, we show an illustrative example of the existence of such skewed triangles in the SfM problem on Alamo, which is a part of the 1DSfM dataset. Occurrences of skewed triangles are common in SfM, which is depicted in Fig. 1 of the main paper, where the scatter plots show many triplets having minimum angles of the triangles (represented on the x-axis) close to zero (lines 274-276 of the main paper). We adopt a non-aggressive strategy to prune the network (lines 238-241 of the main paper) where we retain all the edges of the network ($\mathcal{E}_{ret}$), which are a part of non-skewed triangles. So, even if an edge is a part of both non-skewed and skewed triangles, it will be retained. In this process, only those nodes are retained, which are a part of atleast one triplet which belongs to the set of non-skewed triangles (lines 244-248 of the main paper). Thus, relatively few edges are removed compared to the total edges participating in the skewed triangles. In cases where the directions coincide, like KITTI, the network is not parallel rigid, which means no unique solution exists. Thus, translation averaging cannot be used in such scenarios.
>
> 3. Time comparison with brute-force method: The timing of the brute-force method will be added in the supplementary material.
>
> 4. Minimum angle close to $180^{\circ}$: Fig. 1 of the main paper shows the scatter plots on real data, which contains outliers. Some triplets containing outliers do not form a triangle, and directions in such triplets are far away from being on a plane. Such triplets do not satisfy the constraint of a triangle, i.e. the sum of angles equal to $180^{\circ}$.
>
> 5. Visualizing Eqn. 2: In Fig. R1 (in the rebuttal pdf), we show an intuitive explanation of the conditioning of a triangle under different scenarios. For a well-conditioned triangle, where no angle is close to zero, a small change in direction (green to red) leads to a small change in the absolute translation. But for an ill-conditioned triangle, where atleast one angle is close to zero, a small change in direction leads to a large change in the absolute translation.
>
> 6. Removal of skewed triangles: An edge can participate in more than one triplet. When a triangle is identified as skewed, it is marked for removal. Specifically, identifying which edges are leading to a skewed triangle can be time consuming. We adopt an efficient approach where we classify triangles as skewed or non-skewed. Since an edge can participate in both non-skewed and skewed triangles, the intersection of edges participating in the two classes of triangles is not empty. Now, we can remove the edges either participating in skewed triangles (aggressive) or retain the edges participating in non-skewed triangles (non-aggressive) (lines 238-241 of the main paper). We employ the non-aggressive removal of edges (lines 244-248 of the main paper). So, in effect, we only delete edges without explicitly maintaining a set of edges participating in skewed triangles and do not delete all the edges of the skewed triangles.

---

> > ### Comment · Reviewer_Jy8r · 2023-08-17
> >
> > I appreciate the authors' answers. However, I am still not too convinced by the experiments. There are many other sequential datasets that the authors could use. For example, on EuRoC, global methods tend to work reasonably well. Also, there the camera poses could be directly compared to a GT and not only to a COLMAP reconstruction that might fail in some cases.

---

> > > ### Author Response · Authors · 2023-08-20
> > >
> > > Thank you for the comment. We could not find any resource which shows results using translation averaging on sequential datasets, including EuRoC. So, we tested translation averaging using BATA [56] on the EuRoC MAV dataset and also tried our method on it, following the same experimental procedure as described in Sec. 6 of the main paper. For the "Machine Hall 01" sequence, the mean and RMS errors without our filter are 1.62 m and 1.72 m, respectively, while after using our filter, they reduced to 1.09 m and 1.42 m, respectively. For the "Vicon Room 1 02" sequence, the mean and RMS errors without our filter are 0.79 m and 0.88 m, respectively, and after using our filter, they reduced to 0.67 m and 0.77 m, respectively. Although there is improvement using our filter, the errors in camera trajectories both before and after applying our filter are very high compared to the incremental methods used in SLAM, where the error magnitude is of the order of 0.01 m (ORB-SLAM2 [Mur-Artal \textit{et al.}, 2017]). Moreover, after visual inspection, we found that the trajectories look very different when estimated from translation averaging compared to the provided ground truth. These indicate that the scale estimation in sequential datasets is intrinsically complex using global translation averaging methods.

---

> > > > ### Comment · Reviewer_Jy8r · 2023-08-21
> > > >
> > > > These errors seem indeed a little large. However, I see that the proposed method improves the results here. I am willing to improve my rating to borderline accept since I still believe the technique is a valuable step to having a practical global SfM pipeline.

---

### Official Review · Reviewer_vxHs · 2023-07-02

**Soundness:** 4 excellent
**Presentation:** 3 good
**Contribution:** 3 good
**Rating:** 6
**Confidence:** 5

**Summary:**

This paper analyzed the sensitivity problem in translation averaging. Built upon the theoretical analysis of the skew triangles in a bearing network, this paper also proposes an efficient algorithm to identify and remove edges that can make the translation averaging problem ill-conditioned. The proposed algorithm is integrated into a global SfM pipeline. The experiments are conducted on the 1DSfM dataset, with different translation averaging solvers (RevisedLUD and BATA) used to solve for global positions. The quantitative results show the proposed algorithm effectively reduced the condition number of the angle matrix, and the translation errors are reduced consistently.

**Strengths:**

This paper analyzed the sensitivity problem in translation averaging, which is different from the parallel rigidity problem and has not been considered before. Strict proofs are given for the theorems proposed in the paper. The proposed edge filtering algorithm is very efficient and effective.

**Weaknesses:**

The main limitation is that the analysis and the proposed algorithm are based on the triplets in the graph, where the requirements are not always held in the real world. The sensitivity analysis in the translation averaging problem is useful, however, is less important than the parallel rigidity problem in my point of view. The paper gives sufficient quantitative results to support their method but is lack qualitative results.

**Questions:**

I have no more questions regarding the details of the proposed method. My question is on the experiment part. The authors evaluated their method on the 1DSfM dataset, which has been always used in many structure-from-motion methods. However, I think the dataset is out of date though it is very popular a decade ago. I would like to know did the authors evaluate their method on other more realistic datasets, such as some SLAM datasets and aerial datasets.

**Limitations:**

The limitation is that the proposed algorithm is based on the triplets in the graph. I would like to see an improved version of this paper can be extended to more general graphs. For now, I have to say the method contributes not significantly to the NeurIPS community since it has limited application and is less important than the parallel rigidity analysis.

---

> ### Author Rebuttal · Authors · 2023-08-09
>
> 1. Dataset for evaluation: We use 1DSfM dataset for evaluation. We recompile the input and the ground truth using Colmap (lines 251-254 of the main paper) to get a more reliable reconstruction than the one provided using Bundler. For any dataset which is sequential in nature, like SLAM or aerial datasets, the parallel rigidity of the network is not maintained in general. For instance, estimating scales during colinear translations in SLAM is not feasible based solely on directions as input since infinitely many solutions exist, implying that the network is not parallel rigid and, thus, translation averaging cannot be applied. Also, if the parallel rigid network is extracted in such datasets, the maximal network obtained will be very small compared to the full network, which will make the problem unrealistic.
>
> 2. Analysis based on triplets of graph: In general graphs, many different structures occur in 2D and 3D (apart from triangles and structures derived from triangles), each needing an individual analysis in terms of conditioning of the problem. Also, once the properties of each of those structures are known, it will be a combinatorial problem to analyze the sensitivity of the graph based on these structures since general graphs will be a combination of those structures. Considering triplets of a network has many advantages. At first, we only get a single structure i.e. a triangle which can be used as a building block of the network. Next, it is relatively easy to analyze the sensitivity of a triangle since it is planar (Thms. 1 and 2) and then extend it to a triplet network (Thm. 3). Lastly, we can ensure parallel rigidity based only on the network structure (Thm. 4), making it efficient for practical usage. We take a first step to deal with the issue of sensitivity in translation averaging, and using triplet graphs for analysis has shown many insights into the problem.
>
> 3. Importance of sensitivity analysis: Sensitivity analysis deals with the reliability of the solution based on the input directions, while parallel rigidity addresses an important issue of the uniqueness of the solution. Sensitivity analysis is similar in spirit to the conditioning of a matrix while solving a linear system of equations ($Ax=b$ problem) where the reliability of a solution is studied. Parallel rigidity can be equivalently described with the algebraic rank of a specific matrix [2], which is similar to the analysis of the uniqueness of a solution given a matrix obtained from a linear system of equations. Both are different aspects of translation averaging that need to be taken care of. In our experiments, we have considered only parallel rigid graphs (lines 256-257 of the main paper). In Tables 1 and 3 of the main paper, it can be seen that removing the triangles which are skewed leads to improvements in the translation estimates. Moreover, the nodes which were removed due to the removal of skewed triangles are not estimated properly, as seen in the column ``Removed Node Errors" of the tables. Such filtering also leads to improvement of the reconstructions and faster convergence of bundle adjustment (Table 4 of the main paper). So, we believe that sensitivity analysis is an important aspect of translation averaging, which is as important as parallel rigidity and outlier detection.

---

> > ### Comment · Reviewer_vxHs · 2023-08-20
> > **Thanks for the rebuttal**
> >
> > Thanks to the authors for the rebuttal. Most of my concerns are addressed. I decide to improve my score. The main paper or supplement should include visualizations and further discussion in the rebuttal.

---

> > > ### Author Response · Authors · 2023-08-20
> > >
> > > Thank you for the comment. We will incorporate the visualizations and further discussion in the main paper and supplementary material.

---

### Official Review · Reviewer_Ah44 · 2023-07-06

**Soundness:** 4 excellent
**Presentation:** 3 good
**Contribution:** 3 good
**Rating:** 7
**Confidence:** 3

**Summary:**

The translation averaging problem is considered, i.e recover absolute translations from pairwise relative translation directions. The paper focuses on analyzing the change in solution with small changes in the input relative directions. The smallest problem (3 nodes) is initially considered which allows to understand that skewed triangles (triplets) are problematic. In order to move to a general problem, the authors propose to consider the case of a network containing only triplets. It allows them to defined the conditioning of the translation averaging problem as the condition number of the "angle matrix". They prove a problem is well conditioned if the minimum angle in all triangles are sufficiently large. Thus, they propose a method to remove such triangles. They experimentally demonstrate that including the proposed method within a classical sfm pipeline allows to obtain better absolute translations, as well as more 3D points triangulated and faster BA.

**Strengths:**

1. Analyzing the sensitivity is a non-trivial problem.
2. Several theorems are proven that allow to identify skewed triangles as problematic.
3. The theorems are important to motivate the proposed skewed-triangle-removal algorithm.
4. The experiments consider both the case of outlier free data and real data.
5. The paper is well written and easy to read.

**Weaknesses:**

The proposed algorithm consists in removing information to obtain a problem that is better conditioned. But it is said in the experiments that the skewness of a triangle is not related to the presence of outliers. So if I understood correctly, the removed triangles are not necessarily outliers and thus may contain important information. Thus the proposed algorithm may harm the final reconstruction. Could you please comment on this?

**Questions:**

Please see "weaknesses"

**Limitations:**

.

---

> ### Author Rebuttal · Authors · 2023-08-09
>
> 1. Important information in removed triangles: The triangles which are filtered using our method are not necessarily outliers. From Tables 1 and 3 of the main paper, it can be seen that the error of the removed nodes are high (Removed Node Errors column) compared to the errors of all the cameras, even if there is no outlier (Table 1), which indicates that the removed nodes are not estimated properly. Since the removed nodes are not reliable, removing such nodes improves the reconstruction, which is validated by more points triangulated and fewer iterations of bundle adjustment in Table 4 of the main paper.

---

### Official Review · Reviewer_5RZJ · 2023-07-07

**Soundness:** 2 fair
**Presentation:** 3 good
**Contribution:** 2 fair
**Rating:** 4
**Confidence:** 4

**Summary:**

The authors propose the sensitivity theory for the Translation Average problem (i.e., input is a large number of coordinate vertex point pairs in relative directions observations and the output is the absolute vertex coordinates with consistent scales), which can be used to efficiently identify the inputs that would make the problem ill-condition (i.e., too small relative direction angles and too large uncertainty to form valid constraints) and remove them to improve the overall algorithm accuracy and convergence speed.

**Strengths:**

1.The paper has solid theory, elegant derivation, and creative use of mathematical tools to analyze its uncertainty.
2.The experiments fully confirm the authors’ claim which effectively improve the accuracy of the algorithm and accelerate the convergence speed.

**Weaknesses:**

1.The motivation of this article is not sufficient, I can understand that by comparing the direction of the angle we can determine whether it is an ill condition problem or not, but I do not know whether this situation often occurs in practical applications (from the experimental point of view, the improvement of accuracy is very limited).
2.The article is rather obscure and difficult to understand, especially section4 and section5.
3.Lack of qualitative experiments to demonstrate the importance of such problems and the superiority of the method.

**Questions:**

1.I completely missed the meaning of what is shown in Figure 1, could author reintroduce it
2.Whether this uncertainty can be explicitly modeled as uncertainty of Gaussian distribution and incorporated into the optimization framework for solving absolute vertex positions.

**Limitations:**

1.I think the biggest problem of the paper is that this problem is not a very important problem, first of all, there must be similar methods inside other SFM works to avoid the uncertainty caused by too small parallax. Secondly, whether this extreme case is more common in real scenes is doubtful. Thirdly, if a vertex is observed by more than one other vertex, then the observation in other directions can be used to constraint the absolute vertex solution. So, the authors should explain again why it is an important question which I think may be proved using qualitative experiments.
2.Some parts of the paper are too obscure, it is suggested to add more intuition descriptions, and visual diagrams to aid understanding.

---

> ### Author Rebuttal · Authors · 2023-08-09
>
> 1. Fig. 1 of the main paper: We analyze the real data to understand how frequently skewed triangles occur in real data and whether skewed triangles have any relation to the presence of outliers. For this, we provide scatter plots between the minimum angle in a triangle on the x-axis (which reveals the skewness of the triangle, a smaller minimum angle means a more skewed triangle) and the maximum error of the directions in the triangle (which reveals the presence of outlier, a larger maximum error means the presence of outlier). It can be seen that there are many triplets whose minimum angle is close to zero which reveals that the presence of skewed triangles is common (lines 274-276 of the main paper). Also, the scatter plots imply that there is no relation between the skewness of a triangle and a presence of an outlier in a triangle.
>
> 2. Uncertainty modelling: Given the nature of input (relative directions) and output (absolute translations), modelling uncertainty on absolute translations as Gaussian may not be reflective of the true uncertainty. Even then, if it is modelled as a Gaussian and incorporated into the optimization routine small inaccuracies in the uncertainty estimates can lead to a drastic change in the output when the triangles are skewed, as an implication of Thm. 1, which is undesirable.
>
> 3. Small parallax and occurrence of skewed triangles: Although small parallax is handled in SfM pipelines which could lead to skewed triangles, the existence of skewed triangles is not limited to small parallax. As shown in Fig. R2 (in rebuttal pdf), even cameras with large baselines can lead to skewed triangles. Fig. 1 (of the main paper) also discloses the same where there are many triplets having the minimum angle of the triangle close to zero, denoting the existence of skewed triangles (lines 274-276 of the main paper).
>
> 4. Vertex constraint: If a vertex is constrained by atleast two other vertices, which is always the case for parallel rigid networks, we can get a perfect solution when there is no noise in the directions and no numerical errors. But, in practice, the directions are noisy and rounding-off or truncation errors during computation can lead to errors in the solution of the vertex. This is similar in spirit to the conditioning of a matrix while solving a linear system of equations ($Ax=b$ problem), where numerical issues arise when a matrix is not well-conditioned. Thus, constraining the vertex is not adequate to obtain a reliable solution.
>
> 5. Intuitive descriptions: Fig. R1 (in rebuttal pdf) shows the conditioning of the triangle under different scenarios, which provides examples of well-conditioned and ill-conditioned triangles. Fig. R2 shows an illustrative example of the existence of such skewed triangles by considering the Alamo dataset, where two different cases of ill-conditioned triangles (one small angle and two small angles in a triangle) are shown.
>
> 6. Qualitative results: In Figs. R3 and R4 of the rebuttal pdf, we provide some qualitative results. In Fig. R3, the arch in the Notre Dame reconstruction is misplaced due to improper translation estimates, as seen in the lateral view. Using our filter removes such a misplaced arch. In Fig. R4, the wall in the Piazza del Popolo reconstruction is misplaced, but such an effect is not seen after applying our filter. This reveals that removing skewed triangles improves the reconstruction quality.

---

> > ### Comment · Reviewer_Jy8r · 2023-08-17
> >
> > "I think the biggest problem of the paper is that this problem is not a very important problem, first of all, there must be similar methods inside other SFM works to avoid the uncertainty caused by too small parallax. Secondly, whether this extreme case is more common in real scenes is doubtful."
> >
> > The reviewer is mistaken. The fact that translation averaging is extremely challenging is the reason why global Structure-from-Motion methods are not really used in practice. They are not as good as incremental approaches due to the complexity of translation averaging (i.e., the missing scale information makes the problem very hard). Solving translation averaging accurately would unlock practical global SfM pipelines (instead of incremental ones) reducing the 3D reconstruction processing time by orders of magnitudes. The paper is a new (however, small) step towards this.

---

### Author Rebuttal · Authors · 2023-08-09

We thank the reviewers for their comments. In this section, we provide descriptions for the table and figures presented in the rebuttal pdf (which have the prefix ``R" in their enumeration) and address individual concerns in the individual rebuttal sections.

1. Results with 1DSfM filter (Table R1): Our method removes skewed triangles, while 1DSfM is designed to remove outlier edges, both of which are two distinct aspects of the problem. We apply the 1DSfM filter on real data and then compare without and with our filter in Table R1 using BATA [56]. It can be seen that after applying the 1DSfM filter and then our filter, the translations are estimated in a better manner.  Moreover, our filter + 1DSfM filter removes the nodes which are poorly estimated than only 1DSfM filter, as seen from removed node errors in the unfiltered network in Table R1. Since both outliers and skewed triangles are different issues, combining filters for both types improves accuracy.

2. Practical applicability (Fig. 1 in main paper and Figs. R1, R2): In Fig. R1, we show conditioning of the triangle under different scenarios, with green and red depicting the unperturbed and perturbed directions, respectively. For a well-conditioned triangle, a small change in direction leads to a small change in the absolute translation. But for an ill-conditioned triangle, a small change in the direction leads to a large change in the absolute translation. Fig. R1(a) shows a well-conditioned triangle, Fig R1(b) shows a triangle with one small angle due to which it is ill-conditioned and Fig R1(c) shows a triangle with two small angles making it ill-conditioned (these conditions are also inferred by Thm. 1). Fig. R2 shows an illustrative example of the existence of such skewed triangles in the SfM problem. For the Alamo dataset, most images capture the front part of the museum, and thus, the cameras are densely connected in the network. The BLUE triangle depicts a triplet which is Type-I ill-conditioned triangle and the GREEN triangle shows Type-II ill-conditioned triangle. In Fig. 1 of the main paper, it can be seen from the scatter plots that there are many triplets whose minimum angle (represented on x-axis) is close to zero (lines 274-276 of the main paper). This reveals that the existence of skewed triangles is very common in unordered image collections and thus affects the ability to carry out accurate reconstructions.

3. Qualitative results (Figs. R3, R4): We provide some qualitative results in the rebuttal pdf. In Fig. R3, the arch in the Notre Dame reconstruction is misplaced due to improper translation estimates, as seen in the lateral view. Using our filter removes such a misplaced arch. In Fig. R4, the wall in the Piazza del Popolo reconstruction is misplaced, but such an effect is not seen after applying our filter. This reveals that removing skewed triangles helps get more accurate reconstructions, which is also revealed by more points triangulated for most datasets, as shown in Table 4 of the main paper.

---

### Decision · Program_Chairs · 2023-09-21

**Decision:**

Accept (poster)

**Comment:**

This submission received the scores BR, A, WA, BA

All the reviewers are impressed by the theory.

The negative reviewer (BR) proposed to reject the paper because s/he thought the problem was not important, however the other reviewers thought otherwise.

Two reviewers (WA, BA) had some concerns about the experiments even after rebuttal, however their concerns are minor enough for them to still recommend acceptance. The authors should make an effort to fix these aspects and to emphasize the importance of the problem, as this will make the paper stronger.